# Determining the adsorption energies of small molecules with the intrinsic properties of adsorbates and substrates

Wang Gao [1✉], Yun Chen [1], Bo Li [1], Shan-Ping Liu[1], Xin Liu [1] & Qing Jiang [1✉]

Adsorption is essential for many processes on surfaces; therefore, an accurate prediction of adsorption properties is demanded from both fundamental and technological points of view. Particularly, identifying the intrinsic determinants of adsorption energy has been a long-term goal in surface science. Herein, we propose a predictive model for quantitative determination of the adsorption energies of small molecules on metallic materials and oxides, by using a linear combination of the valence and electronegativity of surface atoms and the coordination of active sites, with the corresponding prefactors determined by the valence of adsorbates. This model quantifies the effect of the intrinsic properties of adsorbates and substrates on adsorbate–substrate bonding, derives naturally the well-known adsorption-energy scaling relations, and accounts for the efficiency and limitation of engineering the adsorption energy and reaction energy. All involved parameters are predictable and thus allow the rapid rational design of materials with optimal adsorption properties.

[1] Key Laboratory of Automobile Materials, Ministry of Education, and College of Materials Science and Engineering, Jilin University, Changchun 130022, China. ✉email: wgao@jlu.edu.cn; jiangq@jlu.edu.cn

The adsorption of a molecule to a solid surface is a general phenomenon in many processes, i.e., heterogeneous catalysis, gas sensors, molecular electronics, biomedical applications, and so on[1–4]. The formation and breaking of chemical bonds are ubiquitous during the processes, whereas the adsorption energy is essential in determining the underlying mechanism. Therefore, finding the physical and chemical determinants of adsorption energy has been one of central goals in the fields[5]. Even with the rapid development of surface science techniques and theoretical methods such as density functional theory (DFT)[6], it remains a formidable challenge to screen rapidly the adsorption energy of all potentially interesting materials. Therefore, it is critical to correlate the intrinsic properties of substrates and adsorbates with the adsorption strength, which can not only identify the fundamental determinants of adsorption energy but also pave the ways of fast estimate of adsorption energy.

Intuitively, the adsorption energy should be a function of the electronic and geometric properties of adsorbates and substrates. A remarkable example with the electronic properties of substrates is the *d*-band center model by Nørskov et al.[2,7–9], which has been particularly successful in elucidating trends in adsorption at surfaces of pure transition metals (TMs), in particular late TMs, and some alloys[9–12]. The upper edge of the *d*-band was further identified to be an improved descriptor for almost all pristine TMs[10]. Moreover, the *d*-band model has served as a basis for understanding the relations between adsorption energies and reaction energetics in experiments and theory[9,11–21]. Accordingly, the model of generalized coordination number ($\overline{CN}$), by Calle-Vallejo et al.[22,23], has been effective in describing the geometric effect of pure metals for adsorption and catalysis, providing rational guidance for engineering the surface structures[24–28]. Thus, it is promising for the material design if one could combine the advantage of the *d*-band model in outlining optimal electronic properties and the $\overline{CN}$ model in finding optimal surface sites. Another breakthrough is the discovery of linear relations of adsorption energies, by Nørskov et al.[11], for atoms and their partially hydrogenated species on TM surfaces. These linear scaling relationships (LSRs), which have been successfully extended to intermetallics[29,30], nanoparticles (NPs)[31], TM compounds[32], etc.[33], not only allow the ascertainment of the trend of catalytic activity but also impose a thermodynamic limitation on some catalytic reactions[12–21,34]. The success of LSRs on TMs has been rationalized with the *d*-band model[11], whereas the slope of LSRs is understood and predicable with the electronic factors of adsorbates[11], and the intercept of LSRs is correlated with the surface geometry[35]. Although the LSRs connect the adsorption energies for atoms and their hydrogenated species, there still lacks a formula or a relationship to bridge the gap between the adsorption energies and the easily accessible intrinsic properties of adsorption systems, particularly by means of incorporating the electronic and geometric properties.

Here we provide a far-reaching solution to this issue, by identifying that the three main factors that control the adsorption energies are the valence and electronegativity of surface atoms, the coordination of active sites, and the valence of adsorbates. The established relation, which incorporates the concepts of the known electronic and geometric descriptors[2,7–9,22,23], predicts the adsorption energies in good agreement with the DFT calculations for twenty species on TM surfaces, NPs, intermetallics, and oxides. Remarkably, our scheme naturally deduces the LSRs and thus uncovers a novel physical picture for understanding the adsorption properties. This model not only provides precise rules for designing the materials with target adsorption properties but also enables quick screening of potentially interesting materials since all parameters are predictable.

## Results

**Adsorbates and substrates.** We study the adsorption behaviors of $CH_x$ ($x = 0, 1, 2, 3$), CO, COH, CHO, CHOH, COOH, $NH_x$ ($x = 0, 1, 2$), $NNH_2$, $OH_x$ ($x = 0, 1$), OOH, $OCH_3$, P, F, and Cl on various substrates. The adopted adsorbates cover the main intermediates for many catalytic reactions, such as $CO_2$ reduction reaction ($CO_2RR$), the decomposition and/or oxidation of $CH_4$, $CH_3OH$ and HCOOH, $N_2$ reduction reaction (NRR), oxygen evolution reaction, and oxygen reduction reaction (ORR). For substrates, we choose a series of TM extended surfaces, NPs, intermetallics, and oxides. TM extended surfaces and NPs have the generalized coordination number $\overline{CN}$ of $1 \sim 8$ (see Supplementary Tables 1 and 2)[22,23]. For a given atom on surface, $\overline{CN}$ is the sum of the weights of its nearest neighbors (that is obtained by dividing their own usual coordination number CN with the usual CN in bulk), whereas the usual CN is the number of its nearest neighbors. The extended surfaces contain the low-index surfaces (111), (100), and (110), cavity (111), stepped surfaces with (211), (532), (553), and (711) steps, kinks, and surfaces with metal adatoms, whereas the NPs include $NP_{13}$, $NP_{38}$, $NP_{44}$, $NP_{46}$, $NP_{47}$, $NP_{49}$, $NP_{68}$, $NP_{79}$, $NP_{120}$, $NP_{147}$, $NP_{171}$, $NP_{181}$, $NP_{201}$, $NP_{309}$, $NP_{365}$, $NP_{586}$, a 2 nm NP, and square-, pentagonal-, and hexagonal-cross-section nanowires. Intermetallics contain near-surface alloys (NSAs): A-B@A(111) and A-B@A(100) with Pt and Pd as host A and up to twenty-seven doping TM B. Oxides include monoxides (MO), dioxides ($MO_2$), and perovskite oxides ($ABO_3$) with the surfaces of MO(100), $MO_2$(110), and $ABO_3$(100). The wide choice of adsorbates and substrates allows us to propose a general framework for understanding the adsorption properties.

**Adsorption model on TMs and NPs.** We first attempt to unravel the role of electronic structure of substrates in determining the adsorption properties on TMs and NPs. Combining a *d*-band model with Muffin-Tin-Orbital theory, one obtains that the contribution of *d*-states to the adsorption strength $E^d$ is proportional to the coupling Hamiltonian matric element $V_{ad}$[36], which is correlated to the spatial extent of the metal *d*-orbital ($r^d$) and the adsorption distance (L), $E^d \propto (V_{ad})^2 \propto (r^d)^3/L^7$. $r^d$ is associated with the *d*-band center or the number of outer electrons[37] and $L$ can be empirically estimated in terms of electronegativity[38]. Therefore, we introduce an electronic descriptor based on the valence and electronegativity ($\chi$) of TMs for describing adsorption properties:

$$\psi = \frac{S_v^2}{\chi^\beta} \qquad (1)$$

where $S_v$ is the number of valence electrons including both *d*- and *s*-electrons. $\beta$ is an index determined by the role of *d*- and *s*-orbitals in valence descriptions and electronegativity: $\beta = 1/2$ for Ag and Au, whereas $\beta = 1$ for the other TMs (*d*- and *s*-orbitals correspond to $\beta = 1/2$, respectively), reflecting the fact that the *d*-state contribution to the adsorbate–surface binding is much less important in Ag and Au than in the other TMs (due to the full-filled *d*-band and the low position of *d*-band center relative to the Fermi levels in Ag and Au)[9,39]. The data of $S_v$, $\chi$, and $\psi$ for TMs are summarized in Supplementary Table 3.

Figure 1 and Supplementary Figs. 1–4 show the adsorption energies as a function of the electronic descriptor $\psi$ for all considered adsorbates. Interestingly, $\psi$ always exhibits linear relations with the adsorption energies regardless of the geometry of substrates[10,11,15,16,18,19,32,37,40–43], as

$$E_{ad} = k\psi + b \qquad (2)$$

Accordingly, we identify that the slope $k$ can be well approximated for the hydrogenates and oxygenates with one kind of

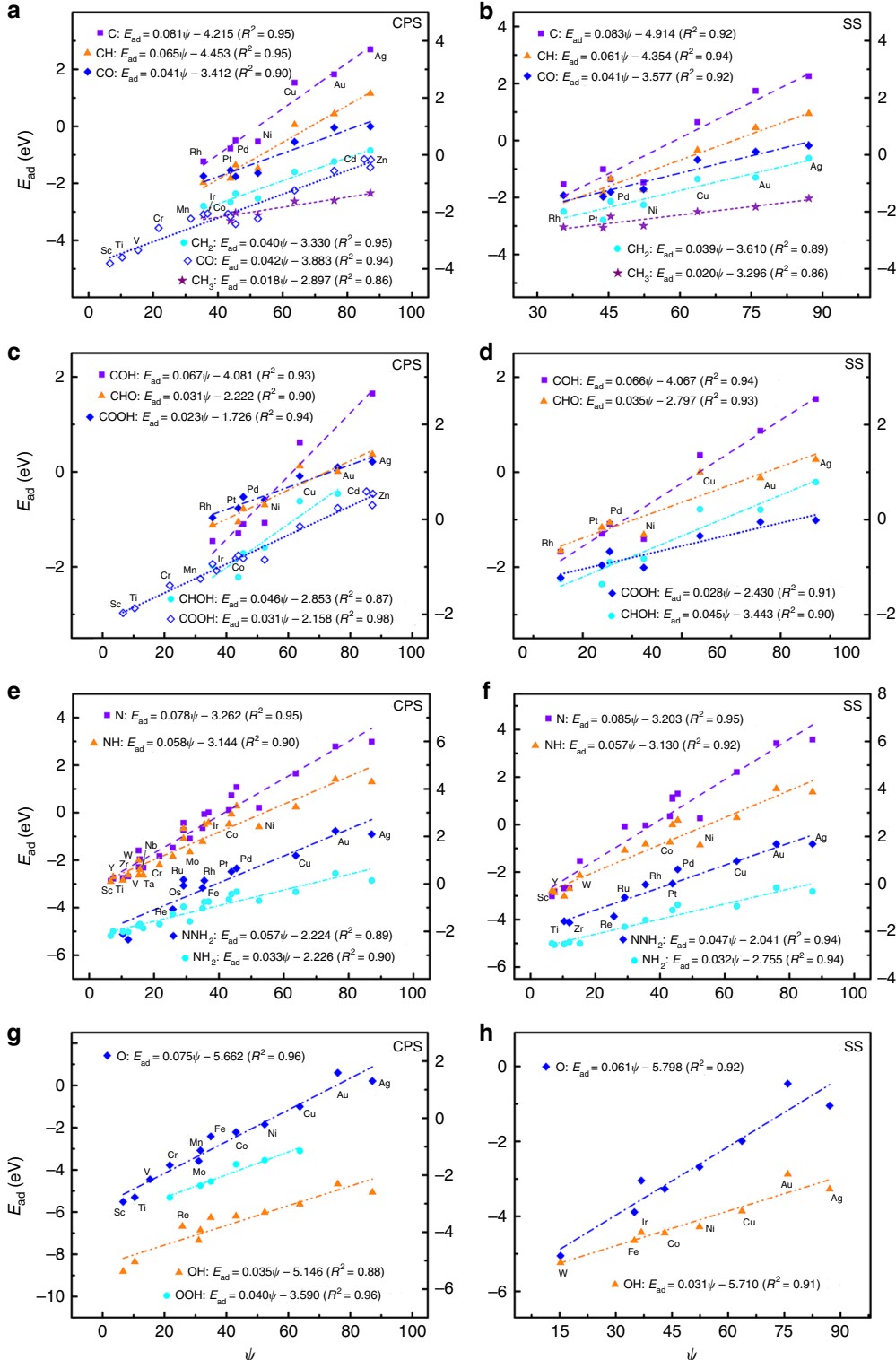

**Fig. 1 Adsorption energies of small molecules on transition metals (TMs) as a function of the electronic descriptor $\psi$. a, b** $CH_x$ and CO on close-packed (**a**) and stepped (**b**) surfaces (CPS and SS) of pure TMs[15]. **c, d** COH, CHO, COOH, and CHOH on CPS (**c**) and SS (**d**) of pure TMs[15]. **e, f** $NH_x$ and $NNH_2$ on CPS (**e**) and SS (**f**) of pure TMs[16]. **g, h** $OH_x$ and OOH on CPS[37,40] (**g**) and SS[11] (**h**) of pure TMs. In each panel, the adsorption energies of adsorbates are linearly correlated with the electronic descriptor $\psi$. The adsorbate name, the color code, and the linear fits are provided as insets and the corresponding parameters are summarized in Supplementary Table 4. In each subfigure with both left and right axes, the linear fits at the upper left corner correspond to the left coordinate axis and those at the bottom right corner correspond to the right coordinate axis. Clearly, the adsorption energy scales linearly with $\psi$, whereas the slope $k$ is governed by electron-counting rules and can be described by the unsaturated bond number of the central atom in adsorbates. Source data are provided as a Source Data file.

functional groups (we will focus on these species, except as otherwise stated):

$$k = 0.1 \times \alpha = 0.1 \times \frac{X_m - X}{X_m + 1} \qquad (3)$$

where $\alpha$ is a characteristic parameter of adsorbates, in which $X_m$ is the maximum number that the central atom of a given adsorbate can bond to the specific coordinating group, and $X$ is correspondingly the actual number. We take a C atom as an example to elucidate the assignment principle for the slope $k$. In the case of $CH_x$, the slope $k$ should be 0.08 for C, 0.06 for CH, 0.04 for $CH_2$, and 0.02 for $CH_3$, because of $X_m = 4$. Considering the hydration reaction in solution, a C atom can bind with five OH, as $HCO_3^-$ ($H_2CO_3 \rightarrow HCO_3^- + H^+$) can be regarded as the stable configuration of $C(OH)_5$, in which the elimination reaction takes place in a pair of OH. The slope $k$ for COH is thus 0.067 with $X_m = 5$. In the case of multi-functional groups, such as COOH, CHO, and CHOH, Eq. (3) can be generalized by including the valence of the second functional group $X'$ with the relation of $\alpha = \frac{X_m - X}{X_m + 1} - \frac{X'}{X'_m + 1}$ (see the details in Supplementary Note 1). The corresponding slopes $k$ of these three species are predicted to be 0.027, 0.03, and 0.047, respectively. The predicted slopes $k$ for all considered

adsorbates are in good agreement with the direct fitting of the DFT calculations (see Fig. 1, Supplementary Figs. 1–4, and Supplementary Tables 4–6), which validates the model for the slope $k$. More importantly, Eqs. (2) and (3) can be used to derive the LSRs of adsorption energies for atoms ($E_{ad}^A$) and their partially hydrogenated species ($E_{ad}^{AH_x}$) automatically, $E_{ad}^{AH_x} = \frac{X_m - X}{X_m} \times E_{ad}^A + \xi$ (see the derivation in Supplementary Note 2)[11]. These results support strongly that our $\psi$-determined scaling model captures the inherent correlation between the adsorption energies and the electronic structures of substrates and adsorbates.

We now turn to study the origin of the offset $b$ for the $\psi$-determined scaling relation, finding that $b$ depends strongly on the geometric structure of substrates and can be characterized as a linear function of the generalized coordination number of surfaces $\overline{CN}$ (see Figs. 2 and 3 and Supplementary Fig. 5)[22,23],

$$b = \lambda \overline{CN} + \theta \qquad (4)$$

with the prefactor $\lambda$ and parameter $\theta$ being constant for a given adsorbate. This correlation is compatible with the findings for the offset of the LSRs on TM surfaces and for the adsorption energies on TM surfaces and NPs[22–28,35,39,44,45]. The underlying mechanism can be attributed to the rule of bond-order conservation[46].

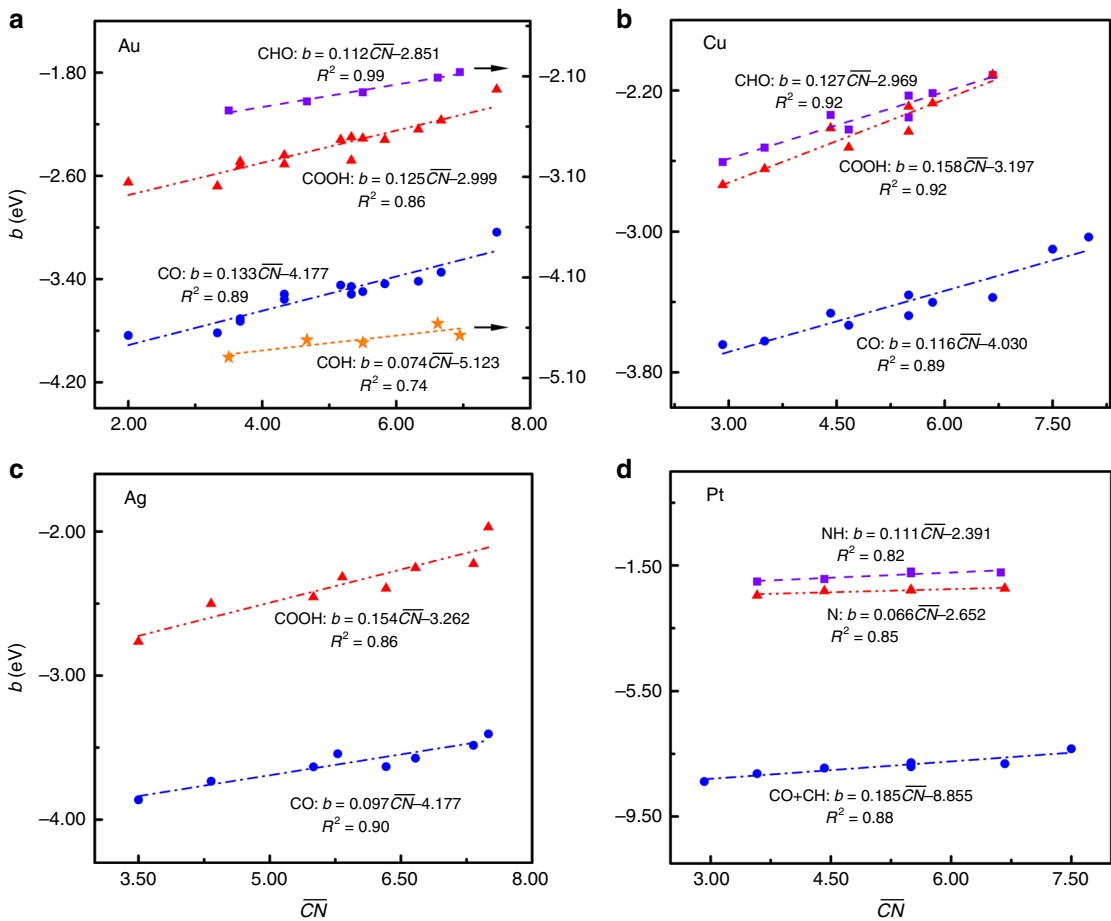

**Fig. 2 Structure-offset $b$ relations for species binding via C and N. a** COOH (red triangles) and CO (blue circles) on fourteen different types of Au surface sites[26], and CHO (purple cubes) and COH (orange star) on five different types of Au surface sites[24]. **b** COOH (red triangles), CO (blue circles), and CHO (purple cubes) on ten different types of Cu surface sites[28]. **c** COOH (red triangles) and CO (blue circles) on eight different types of Ag surface sites[25]. **d** N (purple cubes), NH (red triangles), and CO+CH (blue circles) on seven different types of Pt surface sites[27]. In each panel, the offset $b$ of the $\psi$-determined scaling relation is linearly correlated with the generalized coordination number $\overline{CN}$ of surfaces. The details of surface sites are provided in Supplementary Tables 1, 2, and 7, whereas the linear fits are provided as insets and the corresponding parameters are summarized in Supplementary Table 8. Clearly, the offset $b$ is linearly scaled with the structure parameter following Eq. (4) in the main text. The prefactor $\lambda$ of the term is also governed by electron-counting rules and can be described by the saturated-bond number of the central atom in adsorbates. Source data are provided as a Source Data file.

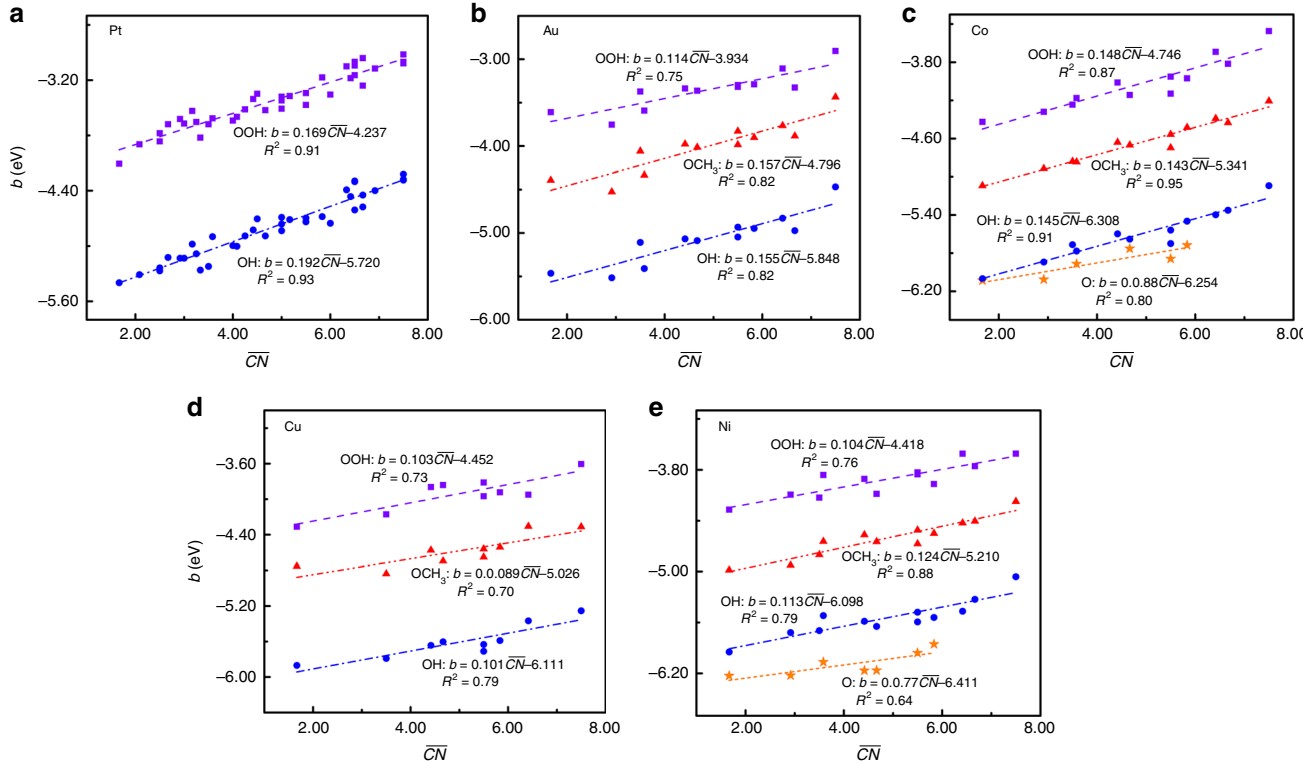

**Fig. 3 Structure-offset $b$ relations for species binding via O. a** OOH (purple cubes) and OH (blue circles) on thirty-eight different types of Pt surface sites[23]. **b–e** OOH (purple cubes), OH (blue circles), OCH$_3$ (red triangles), and O (orange star) on twelve different types of Au (**b**), Co (**c**), Cu (**d**), and Ni (**e**) surface sites[35], where each adsorbate is generally adsorbed on the same site from one surface to the next. In each panel, the offset $b$ of the $\psi$-determined scaling relation is linearly correlated with the generalized coordination number $\overline{CN}$ of surfaces. The details of surface sites are provided in Supplementary Tables 1, 2, and 7, whereas the linear fits are provided as insets and the corresponding parameters are summarized in Supplementary Table 9. Clearly, the offset $b$ is linearly scaled with the structure parameter following Eq. (4) in the main text. The prefactor $\lambda$ of the term is also governed by electron-counting rules and can be described by the saturated-bond number of the central atom in adsorbates. Source data are provided as a Source Data file.

However, the prefactors of the coordination terms for all previously proposed coordination-scaling relations have not been formulated yet[22–28,35,39,44,45]. Herein, we find that the prefactor $\lambda$ in our scheme can be described by the saturated-bond number of the central atom as,

$$\lambda = 0.2 \times (1 - \alpha) = 0.2 \times \frac{X + 1}{X_{\mathbf{m}} + 1} \tag{5}$$

The predictions for the prefactor $\lambda$ by the electron-counting rules are compared with the fitted results from the DFT calculations (see Figs. 2 and 3, and Supplementary Fig. 5 and Supplementary Tables 8 and 9)[22–28,35,39,44,45]. Obviously, the good agreement holds for all adsorbates and substrates under study, which validates the model for the prefactor $\lambda$ (where the constant 0.2 provides a good approximation). With this equation, we can also quantify the effect of adsorbates on the coordination-dependent offset of LSRs ($\xi$)[35], for any pair of adsorbates with the same central atom (see the derivation in Supplementary Note 2), as

$$\xi = 0.2 \times \frac{\alpha_1 - \alpha_2}{\alpha_1} \times \overline{CN} + \theta_{1,2}$$
$$= 0.2 \times \frac{X_2 - X_1}{X_{\mathbf{m}} - X_1} \times \overline{CN} + \theta_{1,2} \tag{6}$$

It is noteworthy that the usual $CN$ is almost identical with $\overline{CN}$ in describing TM surfaces. For the LSRs of O vs. OH, O vs. OOH, and O vs. OCH$_3$, the prefactor $0.2 \times \frac{\alpha_1 - \alpha_2}{\alpha_1}$ is predicted to be ~0.10,

as $X_1 = 0$, $X_2 = 1$, and $X_m = 2$, which is consistent with the fitted data (0.08~0.11) of the DFT calculations[35]. This consistency further confirms the reliability and generality of our scheme.

Thereby we propose the entire expression of the adsorption energy for metallic materials based on the electronic descriptor $\psi$ and the geometric descriptor $\overline{CN}$:

$$E_{\mathrm{ad}} = 0.1 \times \alpha \times \psi + 0.2 \times (1 - \alpha) \times \overline{CN} + \theta$$
$$= 0.1 \times \frac{X_{\mathrm{m}} - X}{X_{\mathrm{m}} + 1} \times \psi + 0.2 \times \frac{X + 1}{X_{\mathrm{m}} + 1} \times \overline{CN} + \theta \tag{7}$$

where the constant $\theta$ is the only parameter that needs to be determined for a given adsorbate, e.g., through DFT calculations, whereas the rest of the parameters are intrinsic and are readily accessible. This enables one to elucidate trends in adsorption at different surfaces rapidly (without DFT calculations), whereas $\theta$ is needed to obtain the values of adsorption energy. $\theta$ most likely originates from the coupling between the states of adsorbates and the $sp$ states of substrates according to the $d$-band model[2,7–9] and is dependent strongly on the bond energy between the atom binding to surface and its coordination atoms in adsorbates (see Supplementary Fig. 6).

**Generalization of the model into intermetallics and oxides.** One of the key advantages of our scheme is that it can be naturally generalized into NSAs and oxides, by including the local environment effect of active centers, such as the effect of the different elements and the local coordination chemistry around

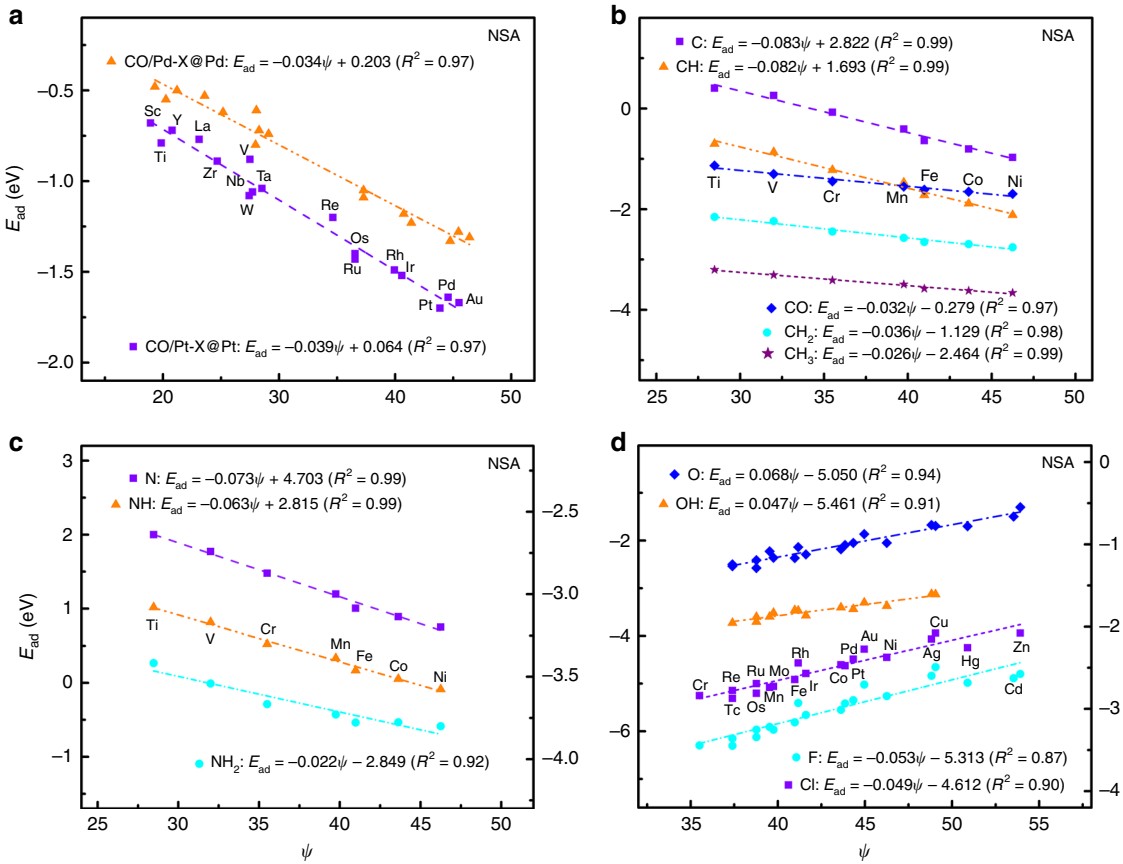

**Fig. 4 Adsorption energies of small molecules on near-surface alloys (NSAs) as a function of the electronic descriptor $\psi$. a** CO on (100) surface of the Pt- and Pd-NSAs[48]. **b–d** CH$_x$ and CO (**b**), N and NH$_x$[47] (**c**), OH$_x$, F, and Cl[30,38] (**d**) on (111) surface of the Pt-NSAs. In each panel, the adsorption energies of adsorbates are linearly correlated with the electronic descriptor $\psi$. The adsorbate name, the color code, and the linear fits are provided as insets and the corresponding parameters are summarized in Supplementary Table 12. In each subfigure with both the left and right axes, the linear fits at the upper left corner correspond to the left coordinate axis and those at the bottom right corner correspond to the right coordinate axis. Clearly, the adsorption energy scales linearly with $\psi$, whereas the slope $k$ is governed by electron-counting rules and can be described by the unsaturated bond number of the central atom in adsorbates. Source data are provided as a Source Data file.

adsorption sites. To do so, the electronic descriptor $\psi$ is generalized by using the geometric mean of the outer-electron number and electronegativity of the given substrate atoms and their neighboring atoms,

$$\psi = \frac{\left(\prod_{i=1}^{N} S_{v_i}\right)^{2/N}}{\left(\prod_{i=1}^{N} \chi_i\right)^{1/N}} \quad (8)$$

where $N$ is the number of the atoms at active centers, whereas $S_{vi}$ and $\chi_i$ are the outer-electron number and electronegativity of the $i$th atom at active centers. See details in Supplementary Note 3 and Supplementary Tables 10 and 11. It is noteworthy that Eq. (8) is automatically converted into Eq. (1) in calculating $\psi$ for pure TMs and NPs, namely Eq. (8) is universal for TMs, NPs, NSAs, and oxides.

We present the adsorption energies of fourteen adsorbates (CH$_x$, CO, NH$_x$, OH$_x$, P, F, and Cl) on the (111) and (100) surfaces of the Pt- and Pd-NSAs[30,38,47,48], and the oxide surfaces MO(100), MO$_2$(110), and ABO$_3$(100)[20,32,37,40] as a function of $\psi$ in Figs. 4 and 5, and Supplementary Figs. 7 and 8. Clearly, $\psi$ always exhibits linear relations with the adsorption energies regardless of NSAs or oxides, and the corresponding slopes are determined by $\frac{X_m - X}{X_m + 1}$. This is particularly encouraging, as the adsorption of OH, F, and Cl on NSAs and the adsorption of OH$_x$ on oxides cannot be accurately described by the $d$-band

model[37,40,47,49]. The slopes (or their absolute values) on NSAs also comply with Eq. (3) as those on TMs and NPs, whereas the minus sign of the slope in Fig. 4a–c reflects that the Pauli repulsion dominates the interaction between the metal $d$-bands and the adsorbate states[47]. Although MO, MO$_2$, and ABO$_3$ oxides have distinct crystal structures, their adsorption energies consistently scale with $\psi$ in an approximate relation of $E_{ad} = \frac{2(X_m - X)}{X_m + 1}\psi + b$ in Fig. 5. The constant 2 is 20 times of that for metallic materials 0.1, ensuring that the efficiency of modulating adsorption energy from one surface to the next is very close on metallic materials and oxides (see Supplementary Tables 3, 10, and 11). All these findings demonstrate that our scheme effectively captures the local environment effects of active centers on NSAs and oxides.

**Mechanistic insights into the established correlation.** We now try to understand the origin of the electronic descriptor $\psi$ by comparing with the previous models. We plot the $d$-band center $\varepsilon_d$ by the Newns–Anderson model[10] (using the semi-elliptical fits to DFT density of state of TMs) with respect to $\psi$ in Fig. 6a (see more in Supplementary Note 4). For the considered twenty-four TMs, there is a clear linear relationship between $\varepsilon_d$ and $\psi$ for seventeen TMs, with the other seven TMs as outliers. Notably, these seven TMs are the ones that have the hybridization energies

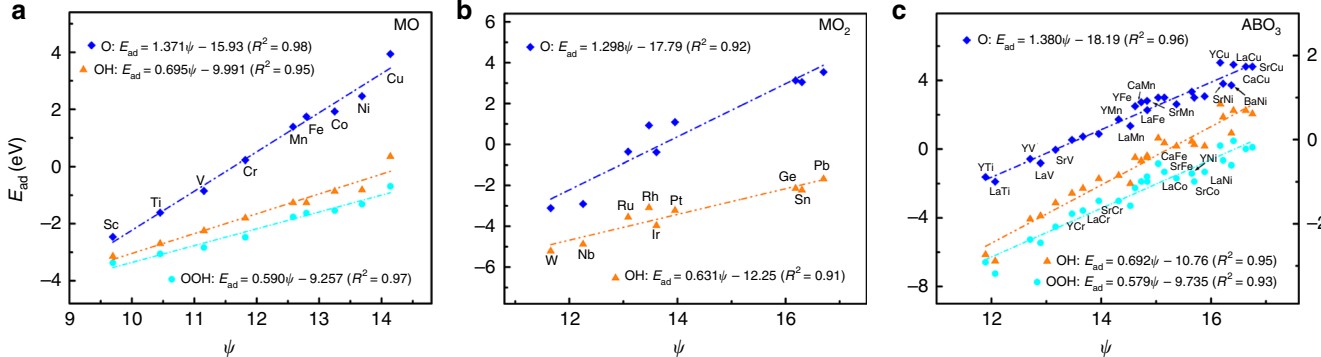

**Fig. 5 Adsorption energies of small molecules on oxides as a function of the electronic descriptor $\psi$. a–c** $OH_x$ and OOH on (100) surface of monoxides MO(100)[37] (**a**), (110) surface of dioxides $MO_2$(110)[40] (**b**), and (100) surface of perovskite oxides $ABO_3$(100)[37] (**c**). In each panel, the adsorption energies of adsorbates are linearly correlated with the electronic descriptor $\psi$. The adsorbate name, the color code, and the linear fits are provided as insets and the corresponding parameters are summarized in Supplementary Table 13. In each subfigure with both the left and right axes, the linear fits at the upper left corner correspond to the left coordinate axis and those at the bottom right corner correspond to the right coordinate axis. Clearly, the adsorption energy scales linearly with $\psi$, whereas the slope $k$ is governed by electron-counting rules and can be described by the unsaturated bond number of the central atom in adsorbates. Source data are provided as a Source Data file.

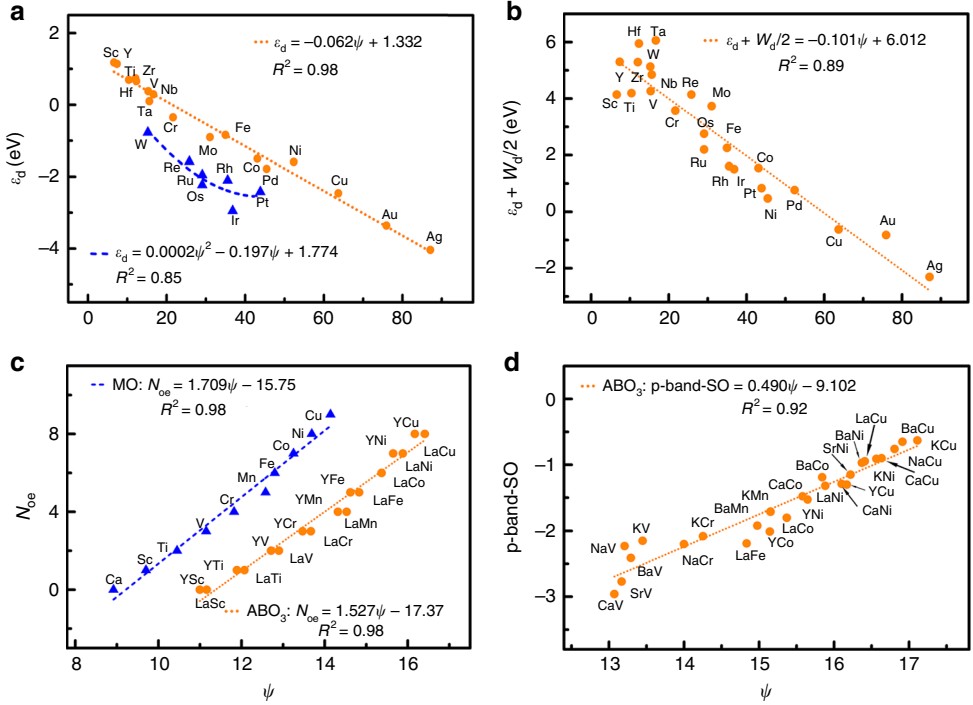

**Fig. 6 Comparison between the electronic descriptor $\psi$ and the previous models on transition metals (TMs) and oxides. a** The $d$-band center $\varepsilon_d$ of the semi-elliptical distribution of density of state (DOS) vs. the electronic descriptor $\psi$[10]. The $\varepsilon_d$ of seventeen TMs is linearly correlated with $\psi$, whereas the $\varepsilon_d$ of the other seven TMs exhibits a quadratic relation with $\psi$. **b** The upper edge of the $d$-states of DFT DOS (namely $d$-band center with a half of $d$-band width, $\varepsilon_d + W_d/2$) vs. the electronic descriptor $\psi$ for the twenty-four TMs[10]. The $\varepsilon_d + W_d/2$ of the twenty-four TMs is linearly correlated with $\psi$. **c** The number of outer electrons $N_{oe}$ vs. the electronic descriptor $\psi$ for monoxides (MO) and perovskite oxides ($ABO_3$)[37]. For both MO and $ABO_3$, the $N_{oe}$ is linearly correlated with $\psi$. **d** The $p$-band center of surface oxygen ($p$-band-SO) vs. the electronic descriptor $\psi$ for $ABO_3$[49]. The $p$-band-SO of $ABO_3$ shows a linear relation with $\psi$. The substrate name, the color code, and the fits are provided as insets. Source data are provided as a Source Data file.

of adsorbate binding depending strongly on $\varepsilon_d$ but weakly on the $d$-band width $W_d$, namely those can be well described by the $d$-band model. By correlating our descriptor $\psi$ with the proposed generalized descriptor $\varepsilon_d + W_d/2$ by DFT[10], namely the upper edge of the $d$-states, we observe a linear relationship for all considered TMs (Fig. 6b). These results suggest that the descriptor $\psi$ essentially reflects the upper edge of the $d$-states and

thus is effective in describing the adsorption energy on TMs and NPs. For the NSAs, $\psi$ likely reflects not only the $d$-band center projected on the adsorption site but also the interaction between the local $d$-states of the substrate and the adsorbate states[47]. In the case of oxides, $\psi$ is correlated linearly with the number of outer electrons $N_{oe}$ and the $p$-band center of surface oxygen ($p$-band-SO) for the considered substrates (Fig. 6c, d)[37,49]. As suggested,

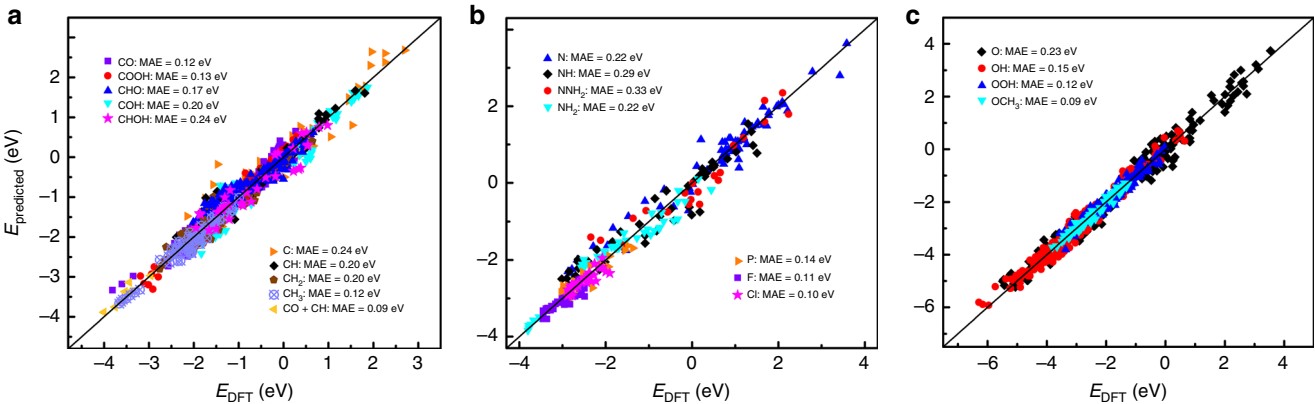

**Fig. 7 Comparison between the predicted adsorption energies and the DFT-calculated results.** In each case, the constant $\theta$ is obtained by using Au data for species binding via C and Cu data for species binding via N and O on transition metals (TMs) and nanoparticles (NPs) (for the few literatures without Au and Cu using the calculated metals), by using Co-doping data (if unavailable using Rh-doping data) on near-surface alloys (NSAs), and by using V-based data (if unavailable using Ir-based data) on oxides. **a**, **b**, and **c** show the results for species binding via C[11,15,18,24–28,30,39,44,45,47,48,51–55] (**a**), species binding via N[16,27,30,47] (**b**) (the relatively large scatter compared with other species mainly due to the change of adsorption sites and adsorption configurations from one surface to the next[16]), and species binding via O[10,11,19,20,22,23,30,32,35,37,38,40–43,49] (**c**). Color code for the type of adsorbates is provided as insets together with mean absolute errors (MAEs). The low MAEs comparable to the approximate error (±0.2 eV) of DFT calculations with (semi-) local functionals, the generality of the studied adsorbates and substrates, and the large amount of available data (2055 different adsorption energies) consistently support the applicability of our scheme and corroborate the proposed physical picture for adsorption as discussed in the text. Therefore, this scheme can be used to predict or estimate the adsorption properties of multifaceted TMs, NSAs, and oxides, with reasonable accuracy. Source data are provided as a Source Data file.

the $N_{oe}$ of metal B is an indicator of its binding ability with oxygen, whereas $p$-band-SO likely reflects the electronic structure perturbation of oxide surfaces induced by adsorption. These results indicate that $\psi$ is linked with the outer-electron characteristics of surface atoms for oxides.

We find that the prefactor of electronic term in our scheme, $\frac{X_m - X}{X_m + 1}$, can be understood or deduced with the effective medium theory (EMT)[50]. In EMT, the adsorption energy of an atom to a metal surface is described as the interaction of this atom with a homogeneous electron gas, making the atom exhibit the electron cloud of a noble gas at the optimal adsorption structures. It is noteworthy that the first-order approximation of the EMT is essential for accurately describing adsorption energies of adsorbates such as oxygen that are not particularly polarizable. The zero-order term of the EMT has been found to generate the relation of $E^{(0)} \propto \frac{(X_m - X)n_0}{X_m}$ in ref. [11] (where $n_0$ is the homogeneous electron density). As the bond number can only be integer in considered adsorbates, we introduce a simple perturbation effect induced by the adsorbed atom to the electron density with $\Delta n = \frac{n_0}{X_m + 1}$, which leads to the first-order term of the EMT with $E^{(1)} \propto \frac{(X_m - X)n_0}{X_m(X_m + 1)} = (X_m - X)n_0 \left[ \frac{1}{X_m} - \frac{1}{X_m + 1} \right]$. As all considered adsorbates are not particularly polarizable, we obtain the relation of $E_{ad} = E^{(0)} - E^{(1)} \propto \frac{(X_m - X)n_0}{X_m + 1}$.

The prefactor of coordination term, $\frac{X + 1}{X_m + 1}$, can be understood in the spirit of bond-order conservation. From the substrate point of view, the adsorption energy is proportional to the coordination of active sites that correspond to the saturated-bond number of active sites. From the adsorbate point of view, this character should hold as well, namely the adsorption energy is proportional to the saturated-bond number of adsorbates, as we found.

## Discussion

Our simple and clear expression of adsorption energy allows one to estimate the adsorption energy of a given species on TM, NP,

NSA, and oxide surfaces given the adsorption energy for just one surface. To demonstrate the accuracy of our scheme, we estimate the adsorption energies for all considered adsorbates and compare the results with the DFT-calculated ones (see Fig. 7). We choose twenty-five TMs with nineteen extended surfaces and another twenty NPs, forty NSAs with two extended surfaces, and eighty-five oxides with three different crystal structures. As the DFT-calculated $\theta$ depends on the specific parameter sets of DFT calculations, it is necessary and coherent to compare the predicted adsorption energies to the calculated ones by the same DFT parameter sets (as those used for obtaining $\theta$). To ensure the consistency of comparison, $\theta$ is obtained with the surfaces that were most calculated in Figs. 1–5 and 7 [10,11,15,16,18–20,22–28,30,32,35,37–45,47–49,51–55]. For instance, we adopt Au for species binding via C and Cu for species binding via N and O on TMs and NPs, because Cu often deviates from the scaling relations for species binding via C and Au is an outlier for species binding via oxygen[35,56–59] (Au is thus used for itself for oxygenates). The mean absolute errors (MAEs) of the predictions relative to the DFT results are ~0.15 eV for species binding via C, ~0.25 eV for species binding via N, ~0.15 eV for species binding via O, and ~0.10 eV for species binding via P, F, and Cl, with most of the deviations less than ±0.2 eV, the approximate error of DFT (semi-)local functionals. This is remarkably encouraging, considering the simplicity of the model, the clear picture of underlying physics, the generality of adsorbates and substrates, and the large amount of available data (2055 different adsorption energies). Although DFT calculations also depend on the electronic and geometric structures of certain materials, the choices of $\theta$ from different materials only have a minor effect on the MAEs of the predictions. When changing $\theta$ from Cu fitted data (−4.452 eV) to Co fitted data (−4.746 eV) gradually for OOH, the MAE is varied by <0.05 eV on TMs and NPs, and by ~0.02 eV on all considered substrates. If using the deviation point Cu to replace Au for determining $\theta$ for species binding via C, the MAE is amplified from ~0.15 eV to ~0.20 eV, which is still within the error of DFT (semi-) local functionals.

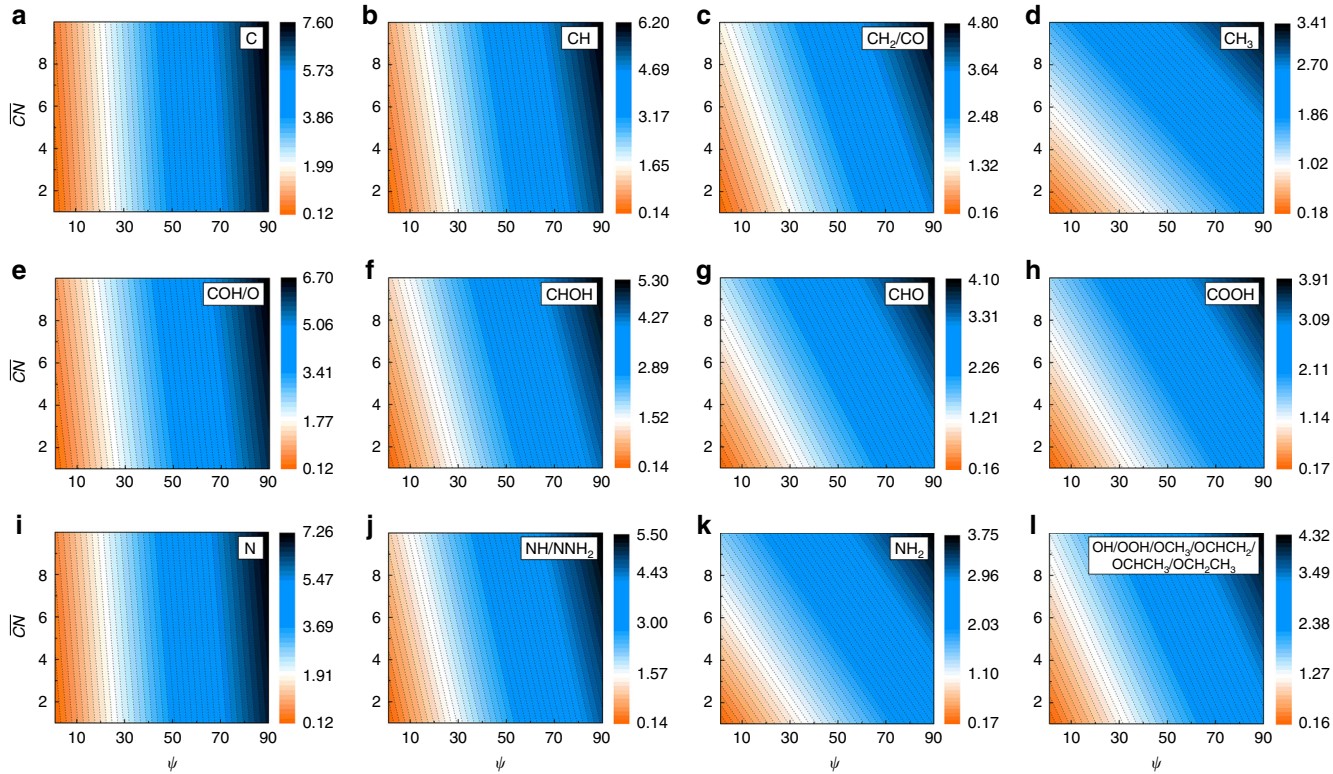

**Fig. 8 Variation of adsorption energy as a function of the electronic descriptor $\psi$ and generalized coordination number $\overline{CN}$. a** C atom. **b** CH radical. **c** $CH_2$ radical or CO molecule. **d** $CH_3$ radical. **e** COH radical or oxygen atom. **f** CHOH radical. **g** CHO radical. **h** COOH radical. **i** N atom. **j** NH radical or $NNH_2$ radical. **k** $NH_2$ radical. **l** OH radical, OOH radical, $OCH_3$ radical, $OCHCH_2$ radical, $OCHCH_3$ radical, or $OCH_2CH_3$ radical. This two-dimensional (2D) mapping is obtained from Eq. (7). The contours and gradients account for the efficiency of modulating the electronic nature and geometric structure of surfaces in engineering the adsorption energy. For modulating the adsorption energy, the electronic descriptor $\psi$ is more effective in unsaturated atoms such as C and N, but the geometric descriptor becomes more important in saturated atoms such as $CH_3$, COOH, and $NH_2$. This 2D mapping can thus serve as a guide for exploring materials with target adsorption properties.

To facilitate the applications of our scheme, we build a two-dimensional mapping of the adsorption energy as a function of $\psi$ and $\overline{CN}$ on TMs and NPs (Fig. 8). We focus on the electronic and geometric terms in Eq. (7), which not only identify the trend of the adsorption energy from one surface to the next but also eliminate the errors of DFT calculations. The contours and gradients in Fig. 8 account for the efficiency of modulating the electronic nature and geometric structure of surfaces in engineering the adsorption energy. Obviously, this modulation efficiency depends strongly on the nature of adsorbates: the electronic descriptor $\psi$ is more effective in unsaturated atoms such as C and N, whereas the geometric descriptor $\overline{CN}$ plays an increasingly important role with increasing the bonding number of the central atom. Our model suggests several NPs and NSAs that exhibit the similar adsorption energy of CO or OH relative to Cu(111) and Pt(111) in Supplementary Table 14. As TMs only exhibit a set of discrete points of $\psi$, NSAs effectively fills in the gaps of $\psi$ between different TMs (see Supplementary Tables 3 and 10), indicating a great potential of alloying in engineering the adsorption energy. It is noteworthy that when modulating $\psi$, metallic materials and oxides are very close in altering the adsorption energy. Clearly, our scheme outlines the perspective for the electronic and geometric properties of adsorbates and substrates in coordinatively engineering the adsorption energies.

Except for engineering the adsorption energies, our scheme also sheds light on the efficiency and limitation of engineering the reaction energies, which can derive naturally the energetic limitations for catalytic reactions. The adsorption energy difference of any pair of adsorbates with the same central atom (a reactant and a product) is as follows:

$$\Delta E_{ad} = 0.1 \times (\alpha_1 - \alpha_2) \times \psi - 0.2 \times (\alpha_1 - \alpha_2) \times \overline{CN} + \theta'_{1,2}$$
$$= 0.1 \times \frac{X_2 - X_1}{X_m + 1} \times \psi - 0.2 \times \frac{X_2 - X_1}{X_m + 1} \times \overline{CN} + \theta'_{1,2} \quad (9)$$

Clearly, the efficiency of $\psi$ and $\overline{CN}$ in engineering the reaction energies is proportional to the bonding number difference of the central atom between a reactant and a product. For $\psi$ and $\overline{CN}$, one is increased and the other one should be decreased, to optimize the modulation efficiency. The predicted prefactors of $\psi$ and $\overline{CN}$ of Eq. (9) are consistent with the direct DFT calculations for $CO_2RR$ (five reaction steps)[15] in Supplementary Table 15. For instance, the predicted prefactors of $\psi$ and $\overline{CN}$ of Eq. (9) for CO→HCO are merely 0.01 and 0.02, respectively, corresponding to a small variation for the energetics of CO→HCO as the nature or structure of surfaces changes. This is in good agreement with the findings in refs. [15,17], where the limiting potential of CO protonation was found to exhibit a nearly horizontal line over TMs. A special case of Eq. (9) appears at $X_1 = X_2$: the adsorption energy difference is always constant for any set of the adsorbates that bind similarly to surfaces regardless of the electronic nature and geometric structure of surfaces (e.g., OH vs. OOH, NH vs. $NNH_2$, and so on in Fig. 8c, e, j, l). This exactly corresponds to the known thermodynamics limitation on ORR[34,37]. Overall, the consistency between our predictions and the literature findings demonstrates the robustness of our model, and further support our model as a guide for the efficient and purposeful design.

We thus suggest improving NRR via the dissociative pathway, as this pathway has the prefactors of $\psi$ and $\overline{CN}$, 0.025 and 0.05, for the energetics of protonation step, whereas the associative pathway comprising NH and $NNH_2$ suffers the thermodynamics limitation.

In summary, we have established a model that determines the adsorption energies by using the valence and electronegativity of surface atoms, the coordination of active sites, and the valence of adsorbates. The electronic descriptor, established from the valence and electronegativity of surface atoms, is powerful, as it effectively reflects the $d$-band characteristics for metallic materials and the outer-electron characteristics of surface atoms for oxides. Our model is descriptive and predictive for twenty species on TMs, NPs, NSAs, and oxides, and can derive automatically the LSRs of the adsorption energies and its generalized form, reflecting the solid physical–chemical basis. This fully predictive scheme uncovers the fundamental physical rules of adsorption, generalizes the efficiency and limitation of engineering the adsorption energy and reaction energy, and allows rapid screening of potentially interesting systems, as all involved parameters are predictable, all of which provide a long-sought guide for future materials design.

## Methods

**Theoretical methods**. In the study, we adopted CASTEP code[60] with ultrasoft pseudopotentials[61] and Perdew−Burke−Ernzerhof (PBE)[62] functional augmented with TSsurf method[63] for all calculations. The TSsurf method has been demonstrated as a reliable method for describing the adsorption structures and the adsorption energies on metal surfaces with the experimental accuracy, because of the inclusion of the screened van der Waals (vdW) interactions. TM surfaces were modeled with four-layer slabs in a unit cell of $p(3 \times 3)$, where the two topmost layers were fully relaxed and the rest of the layers were constrained in the optimized lattice. A vacuum of 24 Å was adopted to separate the adjacent slabs. Careful tests allow us to use plane-wave cutoff energy of 400 eV and the Monkhorst–Pack $k$-point sampling with $2 \times 2 \times 1$ meshes for geometry optimization. The conjugate gradient algorithm was utilized with a convergence threshold of $2.0e^{-5}$ eV and 0.05 eV/Å in Hellmann–Feynman force on each atom. Moreover, the adsorption energy $E_{ad}$ is defined as,

$$E_{ad} = E_{mol/sub} - E_{mol} - E_{sub} \tag{10}$$

where $E_{mol/sub}$ is the total energy of adsorbed system, $E_{mol}$ is the energy of an isolated molecule, and $E_{sub}$ is the energy of clean metal substrate. The gas-phase references are CO, $H_2O$, and $H_2$ for species binding via C, $N_2$, and $H_2$ for species binding via N, and $1/2O_2$, OH, OOH, and $OCH_3$ for O, OH, OOH, and $OCH_3$. It is noteworthy that only the data of CO and COOH adsorption on TMs (hollow diagonal blocks in Fig. 1a, c) were obtained with our PBE + Tssurf calculations, whereas the rest of the data were cited from literatures[10,11,15,16,18–20,22–28,30,32,35,37–45,47–49,51–55]. Thus, the gas-phase references in different literatures need to be unified as shown in Supplementary Note 5. Our calculations demonstrate that the vdW contribution to the adsorption energies of CO and COOH on TM close-packed and stepped surfaces is about 0.2~0.4 eV, shown in Fig. 1a, c.

## Data availability

The source data underlying all Figures and Tables in the main text and Supplementary Information are provided as a Source Data file.

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

## Acknowledgements

We are thankful for the support from the Program of the Thousand Young Talents Plan, the National Natural Science Foundation of China (Numbers 21673095, 11974128, and 51631004), the Opening Project of State Key Laboratory of High Performance Ceramics and Superfine Microstructure (SKL201910SIC), the Program of Innovative Research Team (in Science and Technology) in University of Jilin Province, the Program for JLU (Jilin University) Science and Technology Innovative Research Team (Number 2017TD-09), the Fundamental Research Funds for the Central Universities, and the computing resources of the High Performance Computing Center of Jilin University, China.

## Author contributions

W.G. and Q.J. conceived the original idea and designed the strategy. S.P.L. performed the DFT calculations. W.G. derived the models and analyzed the results. W.G. wrote the manuscript with the contribution from Y.C. B.L. and Y.C. prepared the Supplementary Information. B.L., Y.C., and X.L. drew all figures. All authors have discussed and approved the results and conclusions of this article.

## Competing interests

The authors declare no competing interests.
