## [Peer Review File · Nature Communications]

Reviewers' comments:

Reviewer #1 (Remarks to the Author):

This manuscript describes a model for predicting adsorption energies of adsorbates on transition metals from readily available chemical information. As linear scaling relations between adsorbates are well established in literature, the novel aspect of the work comes from the use of chemical properties to further simplify scaling relations and reduce the computational complexity of obtaining them. Therefore it is critical that this work provide reduced computational complexity, greater insight, and/or more accurate predictions.

It is not clear that this model provides additional insight relative to the well-established Newns-Anderson (NA) Model and linear scaling relations. Previous work has shown that the slope of scaling relations relate to electronic effects and y-intercepts to geometric effects. This work's main insight is to provide a quantitative description of those two effects. However, the authors develop an electronic descriptor that they show to scale linearly with the common NA model descriptor. A constant factor, θ , is introduced to model the y-intercept for each adsorbate. The authors claim this is constant for each adsorbate, however Figure 2 shows this factor changing for a single adsorbate on different surfaces. The authors do develop a novel characteristic parameter, α , to describe the saturation of the adsorbate. However, the main implications of this model reiterate reaction engineering limitations that have been derived from existing linear scaling relations. It is unclear that this model significantly reduces the computational complexity for predicting adsorption energies. A large body of literature has developed scaling relations for every adsorbate-surface pair examined in this study and the extension to less traditional materials, such as oxides and bimetallics, is not shown. The NA model already provides a simple electronic descriptor that can be used to predict adsorption energies from readily accessible material properties. There is also a growing body of machine learning based approaches that predict adsorption energies from material properties at increased accuracy and reduced complexity, albeit with less insight (see ref 1).

Thus, the reviewer does not consider this work presents sufficient insights to be published in Nature Communications.

There are several instances, detailed in other comments below, where it is unclear whether constant factors and mathematical forms of equations were physically justified or fit to the data. It would help understanding to clearly indicate when parameters are chosen for best fit and when they are physically derived.

In Equation (1), the physical explanation for the value of β is not well elaborated. Why is β chosen to be 1/2 and not another value between 0 and 1? Why is an element like copper, which also has a filled d-band like Ag and Au, not chosen to have a reduced electronegativity effect?

In Equation (3), is the denominator chosen to best fit the data?

In Figure 2(e), there is significant scatter for the Oxygen scaling and it is also missing an R2 value. Is there a reason why this scaling appears so anomalous especially relative to other oxygen-bound adsorbates like OH and OOH?

In Figure 2, it is not clear to the reviewer why the scaling lines should be different for different surfaces. Equation (5) shows the prefactor λ as a function of purely adsorbate related quantities and the text indicates that θ is also a constant for a given adsorbate. Thus, it should hold that $(CN)^{-}$ is the only surface-related variable in Equation (4). For example, why does both the slope and y-intercept change for OOH in 4(e)-(f)? Wouldn't the y-intercept changing indicate that θ is changing and so the θ value would really be constant for a given "adsorbate-surface" pair?

Is there any physical meaning to the value of θ ? It would appear to have many systematic trends in Figure 2, such as being larger for more saturated adsorbates.

In Equation (5), the choice of the constant 0.2 factor is not clear. Is this a fitted parameter?

In Figure 3(a), It is not clear why a quadratic fit is used for the outlying materials. Is there a physical justification for this choice of model? Additionally, was the factor of 1/8 for the width of the d-band center a fitted parameter or physically justified?

The authors discussion of the electronic factor's relationship to the Newns-Anderson (NA) model shows that the electronic factor scales linearly with the well-known descriptor, $\epsilon_d + W_d/2$. Since it is well established in literature that $\epsilon_d + W_d/2$ also scales linearly with the adsorption energies shown in Figure 1, it would appear the main value of this new electronic descriptor, ψ , is providing another expression of the NA model descriptor. It is unclear to the reviewer that this new descriptor provides additional physical insight as the descriptor explicitly generated from an appeal to the NA model. Since the d-band center and width are also readily available physical properties, there is no computational complexity reduction as well.

In Figure 4, is there any explanation for why the nitrogen-binding adsorbates have relatively large scatter compared to the carbon and oxygen-binding adsorbates?

On P6, the authors show the value of their model by showing that it can be used to predict the limitations of catalyst engineering. The authors show that the reaction between any two adsorbates would result in a reaction energy which explicitly depended on the relative bond order of the two adsorbates. This equation reduces to the well-known linear scaling relations when similar adsorbates are used. As result many of the electronic insights discussed are already well known in literature.

References

Mie Andersen, Sergey V. Levchenko, Matthias Scheffler, and Karsten Reuter ACS Catalysis 2019 9 (4), 2752-2759

Reviewer #2 (Remarks to the Author):

In this paper, the authors derive a general expression for the adsorption energies of small molecules, hydrogenates and oxygenates, on transition metal surfaces, based on a linear combination of the valence and electronegativity of surface atoms and the coordination of active sites. Such an expression can be very helpful in predicting adsorption energies without performing time-consuming first-principles calculations. Second, it also provides an understanding of the factors that determine adsorption energies. Hence, this is in principle a very interesting and important paper. However, in the present form the paper does not merit publication, in particular as it does not provide proper credit to the work that has been done before.

1) First of all, the title of the paper is much too general. Neither the title nor the abstract reveals that this paper is "only" concerned with the adsorption of small molecules on transition metal surfaces. For example, in catalysis and energy storage, the adsorption of atoms and molecules on oxide surfaces plays a very important role, but this class of adsorption systems is not covered at all in this work. The limitation to adsorption on transition metal surfaces should be reflected in the title and the abstract.

2) In the introduction, the authors claim that "finding the physical and chemical determinants of adsorption energy has been an elusive goal despite many efforts". Now determining the interaction strength from intrinsic properties of the reaction partners, i.e., establishing reactivity concepts,

has been a long-term goal in chemistry. In fact, already in 1981 Fukui and Hoffmann were awarded with the Nobel Prize for developing reactivity concepts, and Hoffmann has extended these concepts also to adsorption on surfaces, see "A chemical and theoretical way to look at bonding on surfaces, Roald Hoffmann, Rev. Mod. Phys. 60, 601 (1988). Furthermore, recently the d-band model has been quite successful in elucidating trends in adsorption at surfaces, which has been supplemented by the concept of generalized coordination numbers by Calle-Vallejo et al.. The author's work in fact incorporates these last two concepts, and they are properly cited, but only later in the text. The authors should properly acknowledge all these important contributions with respect to reactivity concepts already in the introduction, and also acknowledge that they use components of these already existing reactivity concepts for their own work.

3) On page 3 the authors write: "To ensure the generality of substrates, we choose a series of TM extended surfaces and NPs ...". I am not really sure whether this ensures "the generality of substrates", as the authors only look at one particular class of substrates.

4) The authors mention the generalized coordination number at several places in the text and write on page 3: "Note that the usual coordination number CN is almost identical with CN in describing TM surfaces." However, the uninitiated reader will not understand this properly as the authors never explain the difference between generalized CN and usual CN. This should be done in the text.

5) The authors propose to correlate their "descriptor ψ with the proposed generalized descriptor $\epsilon_d + W_d/2$ ", and claim that thus their descriptor "reflects the effect of both ϵ_d and W_d ". However, in the semi-elliptical approximation $\epsilon_d + W_d/2$ exactly corresponds to the upper edge of the d-band, i.e., they replace the d-band center by the upper edge of the d-band. Hence they do not really include the effect of both the d-band center and the band width, but rather introduce a new descriptor. Hence I do not consider it appropriate to write "The electronic descriptor, established from the valence and electronegativity of surface atoms, is powerful since it effectively reflects the effect of both center and width of dband", as the authors do it in the summary. And can the authors provide a rationale why the upper d-band edge corresponds to a good descriptor?

6) On page 4, the authors relate their work to the effective medium theory and restrict the EMT just to the embedding term. Here the authors do not mention that in fact in reference 39 it has been shown that it is necessary to include first- and even second-order correction terms in order to obtain reliable results for adsorption energies.

Review of “Determining the adsorption energies with the intrinsic properties of adsorbates and substrates”

For Nature Communications September 22, 2019

In this paper, predictive equations are proposed for the coefficients of linear scaling laws for adsorption energies in terms of fundamental and easily accessible properties of the adsorbate and substrate (valence and electronegativity of substrate; bonding of the central atom of the adsorbate; and coordination number of the substrate surface). These predictive equations are shown to be very accurate over a wide range of adsorbates and substrates. This work is very significant for insight into the design of surfaces with desired adsorption energies such as in catalyst design. Equations (3), (5), (6 – though I have a question about 6), and finally, therefore, equation (7) are important fundamental and useful observations.

This paper should be accepted for publication after minor revisions because it makes a significant and impactful fundamental contribution that is well carried out and well described. The revisions concern effective and clear communication of the work. I did not identify a scientific limitation.

MINOR REVISIONS TO BE ADDRESSED:

- Figure 1 in the main text and Figures 4 and 5 in the Supplementary Materials are too small to read the text in the legends. Rather than being displayed in two rows of four panels, these figures should be displayed in four rows of two panels, and enlarged accordingly.
- Figure 1 caption: “The element name, the colour code and the linear fits are provided as insets and” -> “The substrate element ordering is identified by labelling some data points. The adsorbate name, the colour code, and the linear fits are provided as insets and”
- page 2, 6 lines from the end: “see the details in Supplementary Note 1”, Do you mean Supplementary Note 2?
- Figure 5: The black text on the coloured background in each panel is hard to read. I suggest white text or back text in a white text box.
- Supplementary Material: I cannot understand where equation (6) comes from. It is stated to come from equations (4) and (5) in the main text, but these equations do not contain ξ . Also, ξ and γ are functions of X, but this dependence is not indicated in the first line of equation (6), nor is it clear which X (X1 or X2) these symbols are associated with. Can you please explain the origins of equation (6) in a little more detail?

GRAMMATICAL SUGGESTIONS:

- Abstract, 1st line: “many processes on surface” -> “many processes on surfaces”
- last paragraph of introduction, 2nd line: “identifying the three main factors” -> “identifying that the three main factors”
- page 2, line 7: “Combining d-band model” -> “Combining a d-band model”

- page 2, 7 lines above equation (1): “extent of metal d-orbital” -> “extent of the metal d-orbital”
- page 2, the line under equation (1): “number of valence electron” -> “number of valence electrons”
- page 2, 9 lines after equation (2): “solution, C atom can bind” -> “solution, a C atom can bind”
- Figure 2 caption: “The prefactor λ of \overline{CN} term” -> “The prefactor λ of the \overline{CN} term”
- page 3, bottom of column 1: “the prefactors of coordination term” -> “the prefactors of the coordination terms”
- page 4, 3 lines under heading Mechanistic insights...: “by NA model” -> “by the NA model”
- page 4, 9 lines from the bottom of column 1: “energies of adsorbates binding” -> “energies of adsorbate binding”
- page 4, 4 lines from the bottom of column 1 AND 7 lines from the bottom of column 1: “If” -> “By”
- page 6, 9 lines above equation (9): should “the rest layers” be “the rest of the layers”?
- Supplementary Figure 2 caption: “using BEEF-vdW functional” -> “using the BEEF-vdW functional”
- Supplementary Figure 3 caption: “using RPBE functional” -> “using the RPBE functional”
- Supplementary Table 3 header: “of periodic table” -> “of the periodic table”
- Supplementary Table 4 header: “of PBE+Tsurf method” -> “of the PBE+Tsurf method”

Response To Reviewers' Comments

Report of Reviewer #1 --

Comments:

This manuscript describes a model for predicting adsorption energies of adsorbates on transition metals from readily available chemical information. As linear scaling relations between adsorbates are well established in literature, the novel aspect of the work comes from the use of chemical properties to further simplify scaling relations and reduce the computational complexity of obtaining them. Therefore it is critical that this work provide reduced computational complexity, greater insight, and/or more accurate predictions.

It is not clear that this model provides additional insight relative to the well-established Newns-Anderson (NA) Model and linear scaling relations. Previous work has shown that the slope of scaling relations relate to electronic effects and y-intercepts to geometric effects. This work's main insight is to provide a quantitative description of those two effects. However, the authors develop an electronic descriptor that they show to scale linearly with the common NA model descriptor. A constant factor, θ , is introduced to model the y-intercept for each adsorbate. The authors claim this is constant for each adsorbate, however Figure 2 shows this factor changing for a single adsorbate on different surfaces. The authors do develop a novel characteristic parameter, α , to describe the saturation of the adsorbate. However, the main implications of this model reiterate reaction engineering limitations that have been derived from existing linear scaling relations.

It is unclear that this model significantly reduces the computational complexity for predicting adsorption energies. A large body of literature has developed scaling relations for every adsorbate-surface pair examined in this study and the extension to less traditional materials, such as oxides and bimetallics, is not shown. The NA model already provides a simple electronic descriptor that can be used to predict adsorption energies from readily accessible material properties. There is also a growing body of machine learning based approaches that predict adsorption energies from material properties at increased accuracy and reduced complexity, albeit with less insight (see ref 1: Mie Andersen, Sergey V. Levchenko, Matthias Scheffler, and Karsten Reuter ACS Catalysis 2019 9 (4), 2752-2759). Thus, the reviewer does not consider this work presents sufficient insights to be published in Nature Communications.

There are several instances, detailed in other comments below, where it is unclear whether constant factors and mathematical forms of equations were physically justified or fit to the data. It would help understanding to clearly indicate when parameters are chosen for best fit and when they are physically derived.

Answer:

Thank the reviewer for showing interest in our work, and highlighting its potential importance. We also thank the reviewer for offering us constructive criticism that helped us improve our manuscript. We regret that our choice of wording in the original submission did not make clear to the reviewer the distinction between this work and previous work. We have improved this aspect in the revised version of the manuscript.

It is known the d-band model by Nørskov *et al.* that describes the electronic effect of substrates has been very successful in elucidating trends in adsorption at transition metals (in particular late transition metals) and some alloys, which has been supplemented by the concept of generalized

coordination numbers by Calle-Vallejo *et al.* that accounts for the geometric effect of metals. Our work in fact incorporates these two concepts with one formula by using the easily accessible intrinsic properties of adsorption systems (as endorsed by Reviewer#2). This allows the rapid positioning of the potential materials with target adsorption properties by directly engineering the electronic and geometric characteristics of the adsorbate and substrate. The well-known linear scaling relations (LSRs in Ref. [11] by Nørskov *et al.* and Ref. [35] by Calle-Vallejo *et al.*) indeed correlate the adsorption energies for atoms and their hydrogenated species, whereas our model bridges the gap between the adsorption energies and the easily accessible intrinsic properties of the adsorbate and substrate: valence and electronegativity of substrate, bonding of the central atom of adsorbate, and coordination number of substrate surface (a new understanding of the factors that determine adsorption energies, as endorsed by Reviewer#2 and Reviewer#3). It is noteworthy that these parameters are easily accessible in the Periodic Table or solid-state tables. Our scheme naturally deduces the LSRs (including quantifying the effect of adsorbates on the coordination-dependent offset of LSRs) and can serve as a reasonable basis to understand the LSRs. Moreover, our model, generalized by including of the local environment effect of active centers (see equation (8) in main text), can describe the adsorption of 20 adsorbates (species binding via C, N, O, P, F and Cl) on transition metals, nanoparticles, near-surface alloys (NSAs), and oxides. In particular, our scheme covers the adsorption of OH, F and Cl on NSAs and the adsorption of OH_x on oxides that cannot be accurately described by the d-band model (Refs [37,40,47,49]). These results strongly support the generality of our scheme.

The main implications of our model are that it can qualitatively analyze the efficiency and limitation of engineering the adsorption energies and reaction energies when modulating the electronic and geometric properties of transition metals, nanoparticles, NSAs, and oxides, which are essentially determined by the properties of adsorbates (see equations (3) and (5) and the related discussions on Page 4). For instance, in modulating the adsorption energies, the electronic properties are more effective in unsaturated atoms like C and N, whereas the geometric properties play an increasingly important role with increasing the bonding number of the central atom. For the modulation of reaction energies, the efficiency is proportional to the bonding number difference of the central atom between a reactant and a product, while the known energetic limitation for catalytic reactions is just one special case of our model. The later consistency further demonstrates the robustness of our model.

We cannot see any contradiction or inappropriateness for the coexistence of the NA model and our model, since correlating the adsorption properties with the electronic and geometric properties of adsorbates and substrates has been one of on-going central goals in the community. The established NA model for the d-band properties, to the best of our knowledge, is mainly applied into pure transition metals, but not into oxides. Besides transition metals and nanoparticles, we have generalized our model into NSAs and oxides, in particular for the adsorption of OH, F and Cl on NSAs and the adsorption of OH_x on oxides that cannot be accurately described by the d-band model (Refs [37,40,47,49]). These results strongly support the generality of our scheme. In addition, the descriptor $\varepsilon_d + W_d/2$ by the NA model was established by making a simple semi-elliptic approximation to the real DFT density of state (DOS) (Ref. [10]), which is still time-consuming. In contrast, the descriptors or the parameters involved in our scheme are in terms of fundamental and easily accessible properties of adsorbates and substrates (without performing time-consuming DFT calculations). Our

scheme thus provides significant insights into the adsorption properties and is predictive over a wide range of adsorbates and substrates.

As stated by the reviewer, machine learning based approaches, which predict the adsorption energies from material properties at increased accuracy and reduced complexity, are generally with less insight. These approaches also rely on the features of substrates and adsorbates, for which the parameters of our scheme could be choices for future studies.

The change of the constant factor, θ , in short, originates from the different ways in calculating the energy references of molecules in different literatures, the error bar of DFT calculations, and the geometric outliers as discussed in Ref. [35]. We have tried to unify the energy references of molecules in the revised manuscript and updated the results, which show that θ is approximately constant for a given adsorbate. For more details please see Answer 4 below and the revision in the updated manuscript.

We have provided more explanations for the constant factors and mathematical forms of equations. Please see the answers below and the revision in the main text.

We agree with the reviewer that our model should not only be applied into transition metals and nanoparticles, and the presentation of our original submission was not optimal. We have generalized our model on transition metals, nanoparticles, NSAs, and oxides and have also improved the presentation substantially to highlight the most important messages. We hope that the revised manuscript will be judged appropriate for publication in *Nature Communications*.

Query 1:

In Equation (1), the physical explanation for the value of β is not well elaborated. Why is β chosen to be 1/2 and not another value between 0 and 1? Why is an element like copper, which also has a filled d-band like Ag and Au, not chosen to have a reduced electronegativity effect?

Answer 1:

We thank the reviewer for the comment. The physical argument is that d-band and s-band have the equal contribution to the value of β : each has 1/2. Since the d-state contribution to the adsorbate-surface binding is much less important in Ag and Au than in the other TMs, β is taken to be 1/2 for Ag and Au. Au and Ag are among the noblest transition metals due to their full-filled d-band and the low position of d-band center relative to the Fermi levels (-3.56 for Au and -4.30 eV for Ag). Although Cu also has full-filled d-band, its d-band center relative to the Fermi level is -2.67 eV, which makes Cu generally much more active than Au and Ag. We added the explanation on Page 4.

Query 2:

In Equation (3), is the denominator chosen to best fit the data?

Answer 2:

We thank the reviewer for the comment. The denominator of equation (3) can be understood or deduced with the effective medium theory (EMT). It is known that the first-order approximation of the EMT is essential for accurately describing adsorption energies of adsorbates like oxygen that are

not particularly polarizable (Ref. [50]). The zero-order term of the EMT generates the relation of $E^{(0)} \propto (X_m - X)n_0/X_m$, while the first-order term of the EMT leads to $E^{(1)} \propto (X_m - X)n_0/[X_m(X_m + 1)]$, where n_0 is the electron density. Therefore, we obtain the relation of $E_{ad} = E^{(0)} - E^{(1)} \propto (X_m - X)/(X_m + 1)$ for all considered adsorbates. Please see Page 6 for details.

Query 3:

In Figure 2(e), there is significant scatter for the Oxygen scaling and it is also missing an R2 value. Is there a reason why this scaling appears so anomalous especially relative to other oxygen-bound adsorbates like OH and OOH?

Answer 3:

We thank the reviewer for bringing up this issue. This is most likely due to the change of adsorption sites for the adsorbates from one surface to the next. The adsorption energies were obtained with the adsorbates binding to the most stable top, bridge or hollow sites (Ref. [35]). OH and OOH molecules are generally adsorbed at top sites, while O atoms can adsorb on top, bridge or hollow sites. The \overline{CN} , the abscissa of Figure 2(g) (namely Figure 2(e) in the original submission), is computed based on the top sites of the different surfaces. Therefore, OH and OOH molecules exhibit better scaling relative to O atoms. We have focused on the adsorption of O atoms with the same sites and show the R^2 values in Figure 2.

Query 4:

In Figure 2, it is not clear to the reviewer why the scaling lines should be different for different surfaces. Equation (5) shows the prefactor λ as a function of purely adsorbate related quantities and the text indicates that θ is also a constant for a given adsorbate. Thus, it should hold that \overline{CN} is the only surface-related variable in Equation (4). For example, why does both the slope and y-intercept change for OOH in 4(e)-(f)? Wouldn't the y-intercept changing indicate that θ is changing and so the θ value would really be constant for a given "adsorbate-surface" pair?

Answer 4:

We thank the reviewer for the comment. The prefactors λ and y-intercepts θ in Figure 2 are the fitted results of the DFT calculations. The differences between the fitted λ and our predictions are mainly due to the error bar of DFT calculations ($\sim \pm 0.2$ eV) and the outliers as discussed in Ref. [35]. For OOH, the slopes are 0.148 on Co surfaces and 0.114 on Au surfaces, both of which are consistent with the predicted value 0.133 of equation (5). As known, Au(100) surface is an outlier from the scaling determined by \overline{CN} (Ref. [35]). If excluding Au(100) surface, the slope is 0.126 on the other Au surfaces for OOH, moving further closer to the predicted value 0.133. The difference of y-intercept for OOH on different TMs is also likely induced by the particularity of Au. Au generally deviates from the ψ scaling determined by up to 0.5 eV (see Figure 1(g)-(h) and Supplementary Figure 4). For Co, Cu, and Ni, the y-intercepts for OOH are essentially identical. For the other adsorbates and substrates in original Figure 2(a)-(d), the difference of y-intercept comes from the different ways in calculating the energy references of molecules in different literatures. We have tried to unify the energy references of molecules in the revised manuscript and demonstrate that the prefactor λ and y-intercept θ are approximately constant for a given adsorbate. Please see Figure 2 and Supplementary Note 4 for the details.

Query 5:

Is there any physical meaning to the value of θ ? It would appear to have many systematic trends in Figure 2, such as being larger for more saturated adsorbates.

Answer 5:

We thank the reviewer for the comment. θ is a constant for a given adsorbate from one surface to the next, and thus it most likely originates from the coupling between the states of adsorbates and the s-states of substrates according to the d-band model. For species binding via oxygen, our results indicate that θ is dependent strongly on the bond energy between oxygen and its coordination atoms in adsorbates (O-O bond for OOH, O-C bond for OCH₃ and O-H bond for OH). Please see the explanation on Page 4 and Supplemental Figure 8.

Query 6:

In Equation (5), the choice of the constant 0.2 factor is not clear. Is this a fitted parameter?

Answer 6:

The constant 0.2 factor, as a fitted parameter, is one of the main observations from Figure 2 (which shows the offset b of the ψ -determined scaling relation plotted against \overline{CN}) and it provides a good description for the prefactor of the \overline{CN} term for the adsorption of 12 adsorbates (species binding via C, N and O) on dozens of transition metals and nanoparticles. We have explained it on Page 4.

Query 7:

In Figure 3(a), It is not clear why a quadratic fit is used for the outlying materials. Is there a physical justification for this choice of model? Additionally, was the factor of 1/8 for the width of the d-band center a fitted parameter or physically justified?

Answer 7:

We thank the reviewer for bringing up this issue. The outlying materials indeed exhibit a quadratic distribution and a quadratic scaling fits these materials better than a linear scaling in Figure 3a (see Supplemental Note 5). The factor of 1/8 for the width of the d-band center is a fitted parameter for elucidating the connection between d-band model and our electronic descriptor.

Query 8:

The authors discussion of the electronic factor's relationship to the Newns-Anderson (NA) model shows that the electronic factor scales linearly with the well-known descriptor, $\varepsilon_d + W_d/2$. Since it is well established in literature that $\varepsilon_d + W_d/2$ also scales linearly with the adsorption energies shown in Figure 1, it would appear the main value of this new electronic descriptor, ψ , is providing another expression of the NA model descriptor. It is unclear to the reviewer that this new descriptor provides additional physical insight as the descriptor explicitly generated from an appeal to the NA model. Since the d-band center and width are also readily available physical properties, there is no computational complexity reduction as well.

Answer 8:

We thank the reviewer for the comment. The descriptor $\varepsilon_d + W_d/2$ by the NA model was established by making a simple semi-elliptic approximation to the real DFT DOS (d-band states) in Ref.[10], which is still time-consuming, while the parameters involved in our scheme are easily accessible in the Periodic Table or solid state tables. The NA model, to the best of our knowledge, is mainly applied

into pure transition metals, but not into oxides. Besides transition metals and nanoparticles, we have generalized our model into NSAs and oxides, in particular for the adsorption of OH, F and Cl on NSAs and the adsorption of OH_x on oxides that cannot be accurately described by the d-band model (Refs [37,40,47,49]). Our method describes the adsorption energies in terms of fundamental and easily accessible properties of adsorbates and substrates, which provides significant insights into the adsorption properties and is predictive over a wide range of adsorbates and substrates. Please see the details in the main text.

Query 9:

In Figure 4, is there any explanation for why the nitrogen-binding adsorbates have relatively large scatter compared to the carbon and oxygen-binding adsorbates?

Answer 9:

We thank the reviewer for the comment. This is mainly due to the change of adsorption sites and adsorption configurations for the nitrogen-binding adsorbates from one surface to the next (Ref. [16]). Taking NNH₂ on close-packed surfaces of transition metals as an example, it adsorbs at hexagonal close-packed (hcp) sites on Ag, Au, Cu, and Re surfaces, at face center cubic (fcc) sites on Pd, Rh, Os, Ti, and Zr surfaces, and at bridge sites on Pt and Ru. In addition, NNH₂ is almost vertical at the fcc sites of Pd and Rh, but is pointing slightly down at the fcc sites of Ti and Zr. We have added the explanation in the caption of Figure 4.

Query 10:

On P6, the authors show the value of their model by showing that it can be used to predict the limitations of catalyst engineering. The authors show that the reaction between any two adsorbates would result in a reaction energy which explicitly depended on the relative bond order of the two adsorbates. This equation reduces to the well-known linear scaling relations when similar adsorbates are used. As result many of the electronic insights discussed are already well known in literature.

Answer 10:

Our model incorporates the concepts of the electronic and geometric descriptors with one formula, bridges the gap between the adsorption energies and the easily accessible intrinsic properties of the adsorbate and substrate, and is predictive over a wide range of adsorbates and substrates. Furthermore, our model can qualitatively analyze the efficiency and limitation of engineering the adsorption energies and reaction energies when modulating the electronic and geometric properties of substrates on transition metals, nanoparticles, NSAs, and oxides, which are essentially determined by the properties of adsorbates (see equations (3) and (5)). For instance, in modulating the adsorption energies, the electronic properties are more effective in unsaturated atoms like C and N, whereas the geometric properties play an increasingly important role with increasing the bonding number of the central atom. For the modulation of reaction energies, the efficiency is proportional to the bonding number difference of the central atom between a reactant and a product, while the known energetic limitation for catalytic reactions is one special case of our model. At this point, the consistency between our prediction and the well-known energetic limitation of catalytic reactions further supports our model. Please see the discussion on Page 7.

Comments:

In this paper, the authors derive a general expression for the adsorption energies of small molecules, hydrogenates and oxygenates, on transition metal surfaces, based on a linear combination of the valence and electronegativity of surface atoms and the coordination of active sites. Such an expression can be very helpful in predicting adsorption energies without performing time-consuming first-principles calculations. Second, it also provides an understanding of the factors that determine adsorption energies. Hence, this is in principle a very interesting and important paper. However, in the present form the paper does not merit publication, in particular as it does not provide proper credit to the work that has been done before.

Answer:

We thank the reviewer very much for the description of our findings, for showing great interest in our work, and for the affirmations on the value and significance of our work. We agree that we touch a topic of interest to a broad community of researchers working on surface science and catalytic chemistry.

We have improved the presentation substantially to acknowledge the important contributions that have been done before. Furthermore, we have generalized our model into NSAs and oxides, by including of the local environment effect of active centers. Now, our model covers the adsorption of 20 adsorbates (three more species P, F and Cl compared to the original submission) on transition metals, nanoparticles, NSAs, and oxides. We hope that the revised manuscript will be judged appropriate for publication in *Nature Communications*.

Query 1:

First of all, the title of the paper is much too general. Neither the title nor the abstract reveals that this paper is 'only' concerned with the adsorption of small molecules on transition metal surfaces. For example, in catalysis and energy storage, the adsorption of atoms and molecules on oxide surfaces plays a very important role, but this class of adsorption systems is not covered at all in this work. The limitation to adsorption on transition metal surfaces should be reflected in the title and the abstract.

Answer 1:

We thank the reviewer for bringing up this issue. We have generalized our model into oxides (monoxides, dioxides, and perovskite oxides) and intermetallics (Pt- and Pd-based NSAs) as well as more molecules (P, F and Cl), some of which cannot be accurately described by the d-band model (Refs [37,40,47,49]). Our model holds for the adsorption of 20 adsorbates (species binding via C, N, O, P, F and Cl) on transition metals, nanoparticles, NSAs and oxides. These results strongly support the generality of our scheme. In addition, we also revised the title and abstract in order to describe our findings more accurately, as the reviewer suggested. Please see the details on Pages 1 and 5.

Query 2:

In the introduction, the authors claim that finding the physical and chemical determinants of adsorption energy has been an elusive goal despite many efforts." Now determining the interaction strength from intrinsic properties of the reaction partners, i.e., establishing reactivity concepts, has been a long-term goal in chemistry. In fact, already in 1981 Fukui and Hoffmann were awarded with

the Nobel Prize for developing reactivity concepts, and Hoffmann has extended these concepts also to adsorption on surfaces, see 'A chemical and theoretical way to look at bonding on surfaces', Roald Hoffmann, Rev. Mod. Phys. 60, 601 (1988). Furthermore, recently the d-band model has been quite successful in elucidating trends in adsorption at surfaces, which has been supplemented by the concept of generalized coordination numbers by Calle-Vallejo et al.. The author's work in fact incorporates these last two concepts, and they are properly cited, but only later in the text. The authors should properly acknowledge all these important contributions with respect to reactivity concepts already in the introduction, and also acknowledge that they use components of these already existing reactivity concepts for their own work.

Answer 2:

We thank the reviewer for bringing up this issue. We have carefully revised the manuscript by rewriting a part of the introduction and acknowledging the previous important contributions. Please see the details on Page 1.

Query 3:

On Page 3 the authors write: 'To ensure the generality of substrates, we choose a series of TM extended surfaces and NPs ...' 'I am not really sure whether this ensures the generality of substrates,' as the authors only look at one particular class of substrates.

Answer 3:

We thank the reviewer for pointing out this issue. We have generalized our model into oxides (monoxides, dioxides, and perovskite oxides) and intermetallics (Pt- and Pd-based NSAs) as well as more adsorbates (P, F and Cl). Our model holds for the adsorption of 20 adsorbates (species binding via C, N, O, P, F and Cl) on transition metals, nanoparticles, NSAs and oxides. These results strongly support the generality of our scheme. We also revised the statement as suggested.

Query 4:

The authors mention the generalized coordination number at several places in the text and write on page 3: 'Note that the usual coordination number CN is almost identical with CN in describing TM surfaces.' 'However, the uninitiated reader will not understand this properly as the authors never explain the difference between generalized CN and usual CN. This should be done in the text.

Answer 4:

We thank the reviewer for the constructive comment. We have provided explanations for the usual coordination number CN and the generalized coordination number \overline{CN} in the revised manuscript on Page 3. For a given atom, the usual CN is the number of its nearest neighbors and the generalized \overline{CN} is the sum of the weights of its nearest neighbors (that is obtained by dividing their own usual CN with the usual CN in bulk).

Query 5:

The authors propose to correlate their descriptor ψ with the proposed generalized descriptor $\epsilon_d + W_d/2$, and claim that thus their descriptor reflects the effect of both ϵ_d and W_d . However, in the semi-elliptical approximation $\epsilon_d + W_d/2$ exactly corresponds to the upper edge of the d-band, i.e., they replace the d-band center by the upper edge of the d-band. Hence they do not really include the effect of both the d-band center and the band width, but rather introduce a new descriptor. Hence I do not consider it appropriate to write 'The electronic descriptor, established from the valence and electronegativity of surface atoms, is powerful since it effectively reflects the

effect of both center and width of d-band," as the authors do it in the summary. And can the authors provide a rationale why the upper d-band edge corresponds to a good descriptor?

Answer 5:

We thank the reviewer for pointing out this issue. We have revised the statements in the updated manuscript. For adsorption on transition metals, the bond strength is given by the position and filling of the anti-bonding states, which are pinned at the upper edge of the d-band. The new descriptor, $\epsilon_d + W_d/2$, accounts for the effects of both the average energy of d-bands and their spread energy on the position of adsorbate-metal antibonding states. We have provided the explanation in Supplemental Note 5.

Query 6:

On page 4, the authors relate their work to the effective medium theory and restrict the EMT just to the embedding term. Here the authors do not mention that in fact in reference 39 it has been shown that it is necessary to include first- and even second-order correction terms in order to obtain reliable results for adsorption energies.

Answer 6:

We thank the reviewer for this important remark. As demonstrated in Ref. [50] (namely Ref. [39] in the original submission), the first-order term is crucial for describing the adsorption energies for adsorbates like oxygen (which is one of the adsorbates in our work), while the second-order term is only important for the particularly polarizable adsorbates like lithium. Now, by further including the first-order term of the EMT, we are able to understand and deduce the prefactor $(X_m - X)/(X_m + 1)$ for the considered adsorbates directly. The zero-order term of the EMT generates the relation of $E^{(0)} \propto (X_m - X)n_0/X_m$, while the first-order term of the EMT leads to $E^{(1)} \propto (X_m - X)n_0/[X_m(X_m + 1)]$, where n_0 is the electron density. Therefore, we obtain the relation of $E_{ad} = E^{(0)} - E^{(1)} \propto (X_m - X)/(X_m + 1)$ for all considered adsorbates (that are not particularly polarizable). See the details on Page 7.

Comments:

In this paper, predictive equations are proposed for the coefficients of linear scaling laws for adsorption energies in terms of fundamental and easily accessible properties of the adsorbate and substrate (valence and electronegativity of substrate; bonding of the central atom of the adsorbate; and coordination number of the substrate surface). These predictive equations are shown to be very accurate over a wide range of adsorbates and substrates. This work is very significant for insight into the design of surfaces with desired adsorption energies such as in catalyst design. Equations (3), (5), (6 –though I have a question about 6), and finally, therefore, equation (7) are important fundamental and useful observations.

This paper should be accepted for publication after minor revisions because it makes a significant and impactful fundamental contribution that is well carried out and well described. The revisions concern effective and clear communication of the work. I did not identify a scientific limitation.

Answer:

We thank the reviewer very much for the detailed description of our findings, for showing great interest in our work, and for the full affirmations on the value and significance of our work. We agree that we touch a topic of interest to a broad community of researchers working on surface science and catalytic chemistry.

Query 1:

Figure 1 in the main text and Figures 4 and 5 in the Supplementary Materials are too small to read the text in the legends. Rather than being displayed in two rows of four panels, these figures should be displayed in four rows of two panels, and enlarged accordingly.

Answer 1:

We thank the reviewer for this suggestion. We have displayed Figure 1 in five rows of three panels, since we have generalized our model into intermetallics (Pt- and Pd-based NSAs) and oxides (monoxides, dioxides, and perovskite oxides) as well as more adsorbates (P, F and Cl). In addition, we have revised Figures 4 and 5 in the Supplementary Materials to two panels (a part of Supplementary Figure 5 has been moved to Figure 2 in the main text). All Figures are enlarged accordingly.

Query 2:

Figure 1 caption: “The element name, the colour code and the linear fits are provided as insets and” -> “The substrate element ordering is identified by labelling some data points. The adsorbate name, the colour code, and the linear fits are provided as insets and”

Answer 2:

We thank the reviewer for this suggestion and have revised the caption of Figure 1 accordingly.

Query 3:

page 2, 6 lines from the end: “see the details in Supplementary Note 1”, Do you mean Supplementary Note 2?

Answer 3:

We thank the reviewer for pointing out this issue. We have revised the Supplementary Materials carefully and updated the sequence of the Notes accordingly.

Query 4:

Figure 5: The black text on the coloured background in each panel is hard to read. I suggest white text or back text in a white text box.

Answer 4:

We thank the reviewer for this suggestion and have revised Figure 5 accordingly, by using black text in a white text box in Figure 5.

Query 5:

Supplementary Material: I cannot understand where equation (6) comes from. It is stated to come from equations (4) and (5) in the main text, but these equations do not contain ζ . Also, ζ and γ are functions of X , but this dependence is not indicated in the first line of equation (6), nor is it clear which X (X_1 or X_2) these symbols are associated with. Can you please explain the origins of equation (6) in a little more detail?

Answer 5:

We thank the reviewer for pointing out this issue. We regret that the description of the derivation process was incomplete in the previous submission. We have provided more details for the derivation of the equations in Supplemental Note 2.

Query 6:

GRAMMATICAL SUGGESTIONS:

- Abstract, 1st line: "many processes on surface" -> "many processes on surfaces"
- last paragraph of introduction, 2nd line: "identifying the three main factors" -> "identifying that the three main factors"
- page 2, line 7: "Combining d-band model" -> "Combining a d-band model"- page 2, 7 lines above equation (1): "extent of metal d-orbital" -> "extent of the metal d-orbital"
- page 2, the line under equation (1): "number of valence electron" -> "number of valence electrons"
- page 2, 9 lines after equation (2): "solution, C atom can bind" -> "solution, a C atom can bind"
- Figure 2 caption: "The prefactor λ of \overline{CN} term" -> "The prefactor λ of the \overline{CN} term"
- page 3, bottom of column 1: "the prefactors of coordination term" -> "the prefactors of the coordination terms"
- page 4, 3 lines under heading Mechanistic insights...: "by NA model" -> "by the NA model"
- page 4, 9 lines from the bottom of column 1: "energies of adsorbates binding" -> "energies of adsorbate binding"
- page 4, 4 lines from the bottom of column 1 AND 7 lines from the bottom of column 1: "If" -> "By"
- page 6, 9 lines above equation (9): should "the rest layers" be "the rest of the layers"?
- Supplementary Figure 2 caption: "using BEEF-vdW functional" -> "using the BEEF-vdW functional"
- Supplementary Figure 3 caption: "using RPBE functional" -> "using the RPBE functional"
- Supplementary Table 3 header: "of periodic table" -> "of the periodic table"
- Supplementary Table 4 header: "of PBE+Tsurf method" -> "of the PBE+Tsurf method"

Answer 6:

We thank the reviewer very much for the suggestions. We have revised the manuscript and the Supplementary Materials according to the suggestions.

Reviewers' comments:

Reviewer #1 (Remarks to the Author):

This manuscript focuses on a model for predicting adsorption energies from intrinsic properties of adsorbates and substrates. From its original submission, the manuscript has been updated in three key ways. First, alloys and oxides have been added to the model to illustrate its generality. Second, prior work has been more directly referenced throughout the paper. Finally, a more in-depth discussion of model outliers has been added. While these updates improve the manuscript, it is not clear that they solve all the major issues outlined in the first referee report.

One outstanding concern is the constant θ . Since this quantity is obtained directly from DFT it is understandable that there will be error in the value obtained from computations. However, if this model is to reduce computational complexity, then the choice of material used to obtain θ should be irrelevant.

1) From Figure 2 (e)-(f) the value for θ for OOH varies from -3.93 eV (Au) to -4.75 eV (Co). Even if gold is omitted as an outlier the difference in θ values is approximately 0.5 eV. This would mean that if Pt's value for θ is used to predict the Co OOH binding energy, the value should be off by approximately 0.5 eV, as Eads is directly correlated with θ . From Figure 4, it appears that this spread in θ values does not result in a similar spread in the predicted adsorption energy as the MAE for OOH is 0.1 eV. It is not clear to the reviewer how this error is prevented from propagating to the adsorption energies. It would improve the manuscript to directly address the variance in θ in the text.

2) From the caption of Figure 4, the authors indicate that multiple θ values are used, either Cu or Au for transition metals. It is not clear why multiple values are used and for which materials the different θ 's are being applied to. Since in practice researchers will not know which material will yield the θ with the smallest error for their particular adsorbate, shouldn't the MAE be calculated with a single θ ? How is the MAE affected when only 1 θ value is allowed?

The other aspect of the manuscript that may need revision is its discussion of the model's computational complexity reduction. Several times throughout the manuscript, such as the final sentence of the abstract and final sentence of the conclusion, the authors state that the model uses "intrinsic and readily accessible" parameters. However, this model does not remove the need for DFT, it simply reduces the number of calculations that need to be done. It should be clear to readers that this model still includes a DFT calculated parameter, θ . Currently, if a researcher wants to understand a reaction on a new class of materials they need to develop linear scaling relation for each adsorbate. This would require ~ 5 DFT surface adsorption energies for each adsorbate on each facet of the surface. If Calle-Vallejo et al. generalized coordination number is used, then only 1 facet needs to be probed. Using the author's new method 1 DFT surface adsorption energy is required per adsorbate. This is a meaningful computational complexity reduction, but it does not completely remove DFT calculations. The text should be updated so that it does not misleadingly state that all parameters of the model are "readily accessible".

Reviewer #2 (Remarks to the Author):

The authors have now extended their study to some more adsorbates, but most importantly also to oxides and near-surface alloys as substrates. They modified the title which is now more appropriate. Furthermore, they have re-written the introduction and now more properly acknowledge previous works with regard to reactivity concepts. They would have made the job of the reviewers much more easier if they had explicitly indicated the changes they had made to the text, the omission of that is not very professional. Still, the revised version of the paper has been improved significantly. It offers a very interesting general expression for adsorption energies of small molecules on various substrates that will be very helpful, but will also stimulate further

discussion about reactivity concepts. Hence this paper now merits publication in Nature Communications in its present form.

Comments:

This manuscript focuses on a model for predicting adsorption energies from intrinsic properties of adsorbates and substrates. From its original submission, the manuscript has been updated in three key ways. First, alloys and oxides have been added to the model to illustrate its generality. Second, prior work has been more directly referenced throughout the paper. Finally, a more in-depth discussion of model outliers has been added. While these updates improve the manuscript, it is not clear that they solve all the major issues outlined in the first referee report. One outstanding concern is the constant θ . Since this quantity is obtained directly from DFT it is understandable that there will be error in the value obtained from computations. However, if this model is to reduce computational complexity, then the choice of material used to obtain θ should be irrelevant.

Answer:

Thank the reviewer very much for highlighting the key improvements of our revised manuscript and showing great interest in our work and also for offering us further constructive criticism that helped us improve our manuscript. We have revised the manuscript carefully according to the suggestions.

We regret that our explanations about the constant θ in the previous submission were incomplete, and have revised this part carefully. Please see the answers below and the revision in the main text. We hope that the revised manuscript will be judged appropriate for publication in *Nature Communications*.

Query 1:

From Figure 2 (e)-(f) the value for θ for OOH varies from -3.93 eV (Au) to -4.75 eV (Co). Even if gold is omitted as an outlier the difference in θ values is approximately 0.5 eV. This would mean that if Pt's value for θ is used to predict the Co OOH binding energy, the value should be off by approximately 0.5 eV, as Eads is directly correlated with θ . From Figure 4, it appears that this spread in θ values does not result in a similar spread in the predicted adsorption energy as the MAE for OOH is 0.1 eV. It is not clear to the reviewer how this error is prevented from propagating to the adsorption energies. It would improve the manuscript to directly address the variance in θ in the text.

Answer 1:

We thank the reviewer for bringing out this issue. For the fitted θ in Figure 2 (e)-(f), the initial adsorption-energy data of Pt are from Ref. [23] and those of Ni, Cu and Co are from Ref. [35]. To eliminate the computation error by the different parameter sets of different literatures, it is necessary and coherent to compare the predicted adsorption energies to the calculated ones by the same DFT parameter sets (as those used for obtaining θ), namely doing a self-consistent comparison. It is thus more reasonable to adopt the data from Ref. [35] to address the variance in θ . We find that the choices of θ from different metals only have a minor effect on the MAEs of the predictions. If one changes θ from Cu's fitted value (-4.452 eV) to Co's fitted value (-4.746 eV) gradually, the MAE is varied by < 0.05 eV for OOH on Ni, Cu and Co and by ~0.02 eV for OOH on all considered substrates. Even if one compulsively describes the data across the different literatures, for example, by using Pt's fitted θ (-4.237 eV) from Ref. [23] to predict the adsorption energy of all cited metals except Au for OOH, the MAE is ~0.23 eV on TMs and NPs and ~0.21 eV on all considered substrates. The small MAE

~0.1 eV for OOH is the result of the self-consistent comparison and the average of a large amount of data on TMs, NPs, and oxides. Please see the details in the main text and the answer to query 2.

Query 2:

From the caption of Figure 4, the authors indicate that multiple θ values are used, either Cu or Au for transition metals. It is not clear why multiple values are used and for which materials the different θ 's are being applied to. Since in practice researchers will not know which material will yield the θ with the smallest error for their particular adsorbate, shouldn't the MAE be calculated with a single θ ? How is the MAE affected when only 1 θ value is allowed?

Answer 2:

We thank the reviewer for the comments. We adopted Cu and Au for determining θ on TMs and NPs, because they were most calculated in Figures 1, 2 and 4, and thus are beneficial to eliminating the material-dependent error of DFT calculations in determining θ when comparing the predicted and calculated adsorption energy. In addition, Cu often deviates from the scaling relations for species binding via C and Au is an outlier for species binding via O. To ensure the consistency of comparison, we now turn to adopt Au for species binding via C and Cu for species binding via N and O on TMs and NPs, while the outlier Au is used for itself for oxygenates. This choice is found to generate the adsorption energy with reasonable accuracy for all considered adsorbates and metals (see Figure 4). The choices of θ from different materials only have a minor effect on the MAEs of the predictions. If using the deviation point Cu to replace Au for determining θ for species binding via C, the MAE is amplified from ~0.15 eV to ~0.20 eV that is still within the error of DFT (semi-)local functionals. Namely, our scheme is effective in predicting adsorption energy regardless of the choices of θ . Please see the revision in the main text (the first paragraph of Discussion) and the caption of Figure 4.

Query 3:

The other aspect of the manuscript that may need revision is its discussion of the model's computational complexity reduction. Several times throughout the manuscript, such as the final sentence of the abstract and final sentence of the conclusion, the authors state that the model uses "intrinsic and readily accessible" parameters. However, this model does not remove the need for DFT, it simply reduces the number of calculations that need to be done. It should be clear to readers that this model still includes a DFT calculated parameter, θ . Currently, if a researcher wants to understand a reaction on a new class of materials they need to develop linear scaling relation for each adsorbate. This would require ~5 DFT surface adsorption energies for each adsorbate on each facet of the surface. If Calle-Vallejo et al. generalized coordination number is used, then only 1 facet needs to be probed. Using the author's new method 1 DFT surface adsorption energy is required per adsorbate. This is a meaningful computational complexity reduction, but it does not completely remove DFT calculations. The text should be updated so that it does not misleadingly state that all parameters of the model are "readily accessible".

Answer 3:

We thank the reviewer for the comments. We regret that our previous statements were misleading. Indeed, the parameters except θ involved in our scheme are intrinsic and readily accessible. This enables one to elucidate trends in adsorption at different surfaces rapidly (without DFT calculations). A DFT calculated parameter, θ , is needed in our scheme in order to obtain the values of adsorption energy. We have revised the statements in the manuscript according to the comments.

Report of Reviewer #2 --

Comments:

The authors have now extended their study to some more adsorbates, but most importantly also to oxides and near-surface alloys as substrates. They modified the title which is now more appropriate. Furthermore, they have re-written the introduction and now more properly acknowledge previous works with regard to reactivity concepts. They would have made the job of the reviewers much more easier if they had explicitly indicated the changes they had made to the text, the omission of that is not very professional. Still, the revised version of the paper has been improved significantly. It offers a very interesting general expression for adsorption energies of small molecules on various substrates that will be very helpful, but will also stimulate further discussion about reactivity concepts. Hence this paper now merits publication in Nature Communications in its present form.

Answer:

We appreciate the reviewer for his/her positive comments and recommendation for publication in Nature Communications.

REVIEWERS' COMMENTS:

Reviewer #1 (Remarks to the Author):

A FURTHER IMPROVED MANUSCRIPT. I HAVE NO MORE COMMENTS.

Report of Reviewer #1 --

Comments:

A FURTHER IMPROVED MANUSCRIPT. I HAVE NO MORE COMMENTS.

Answer:

Thank the reviewer very much for highlighting the key improvements of our revised manuscript and for supporting the publication of our study in *Nature Communications*.